# Typhoid toxin of *Salmonella* Typhi elicits host antimicrobial response during acute typhoid fever

Salma Srour [1], Francesca K Brown [1], James W Sheffield[1], Mohamed ElGhazaly [1,2,3], Daniel O'Connor [4], Malick M Gibani[5], Thomas C Darton[6], Andrew J Pollard[4], Mark O Collins [1] & Daniel Humphreys [1]✉

## Abstract

*Salmonella* Typhi secretes typhoid toxin that activates cellular DNA damage responses (DDR) during acute typhoid fever. Human infection challenge studies revealed that the toxin suppresses bacteraemia via unknown mechanisms. Using quantitative proteomic analysis on the plasma of bacteraemic participants, we demonstrate that wild-type toxigenic *Salmonella* induced secretion of lysozyme (LYZ) and apolipoprotein C3 (APOC3). Recombinant typhoid toxin or *Salmonella* infection recapitulated LYZ and APOC3 secretion in cultured cells, which involved ATM/ATR-dependent DDRs and confirmed observations in typhoid fever. LYZ caused spheroplast formation, inhibited the *Salmonella* type 3 secretion system, and intracellular infections. LYZ expression was regulated by p53 in a cell type-specific manner and driven by mitochondrial oxidative stress that caused nuclear DDRs and p53-mediated senescence responses. Addition of LYZ inhibited oxidative DNA damage and resulting senescence responses caused by typhoid toxin. Our findings may indicate that toxin-induced DDRs elicit antimicrobial responses, which suppress *Salmonella* bacteraemia during typhoid fever.

**Keywords** DNA Damage Responses; *Salmonella* Infection; Senescence; Lysozyme; Bacteraemia
**Subject Category** Microbiology, Virology & Host Pathogen Interaction

## Introduction

Acute typhoid fever is caused by *Salmonella enterica* serovar Typhi (11 million cases, 116,800 deaths per year), which is a major health problem disproportionately affecting low- and middle-income countries (Meiring et al, 2023). Typhoid fever is established when *S*. Typhi invades the intestinal mucosa from where the pathogen disseminates into the bloodstream resulting in an asymptomatic primary bacteraemia. Replication then occurs in lymphoid tissues and, after an incubation period of 7-10 days, a secondary bacteraemia coincides with febrile symptoms and the onset of acute enteric fever and shedding of transmissible bacteria in stool. A small proportion of individuals develop asymptomatic chronic *S*. Typhi carriage, further contributing to ongoing community transmission to new hosts (Meiring et al, 2023). The control of *S*. Typhi is possible through provision of clean water and vaccines, but hampered by inadequate diagnostics, and rising antimicrobial-resistance.

To initiate infections, *Salmonella* inject virulence effectors directly into host intestinal epithelial cells using the *Salmonella* pathogenicity island 1 (SPI-1)-encoded type 3 secretion system (T3SS) that mediates pathogen macropinocytosis (McGhie et al, 2009). This includes essential effectors such as SipB (Kaniga et al, 1995), which form a translocon in the host plasma membrane through which effectors such as SopE, SptP and SopB are translocated to manipulate Rho and Arf GTPase signalling (Hardt et al, 1998; Humphreys et al, 2012; McGhie et al, 2009; Norris et al, 1998; Stebbins and Galan, 2000). Following macropinocytosis, *S*. Typhi resides with a *Salmonella*-containing vacuole (SCV) where SPI-1 T3SS-injected effectors regulate its membrane trafficking while the SPI-2-encoded T3SS injects effectors that are important for intracellular survival (McGhie et al, 2009). In the SCV, *S*. Typhi expresses the typhoid toxin comprising PltB-PltA-CdtB subunits that are exocytosed into the extracellular milieu (Spano et al, 2008). Once deployed, the PltB subunit binds to sialylated glycans on host surface receptors facilitating toxin endocytosis (Song et al, 2013). Reduction of disulphide bonds linking PltA-CdtB liberates the toxigenic DNase1-like subunit CdtB, which translocates to the nucleus where it activates DDRs through nuclease activity (Balsas et al, 2021; Ibler et al, 2019; Song et al, 2013; Spano et al, 2008). Typhoid toxin can also be assembled with a PltC subunit in place of PltB with both toxin variants sharing the same toxigenic subunit CdtB (Fowler et al, 2019). In addition to *S*. Typhi, typhoid toxin is expressed by the typhoidal serovar *S*. Paratyphi A (Song et al, 2013), and ~40 serovars of non-typhoidal *Salmonella* (+2500 serovars), of which the best studied is *S*. Javiana (den Bakker et al, 2011; ElGhazaly et al, 2023; Ibler et al, 2019; Lee et al, 2020; Miller and Wiedmann, 2016).

[1]School of Bioscience, University of Sheffield, Sheffield, UK. [2]Division of Infection and Immunity, University College London, London, UK. [3]Centre for Immunobiology and Infection, Blizard Institute, Queen Mary University of London, London, UK. [4]Oxford Vaccine Group, Department of Paediatrics, University of Oxford, and the NIHR Oxford Biomedical Research Centre, Oxford, UK. [5]Department of Infectious Disease, Imperial College London, London, UK. [6]School of Medicine and Population Health, University of Sheffield, Sheffield, UK. ✉E-mail: d.humphreys@sheffield.ac.uk

The mechanisms by which toxin-mediated host DDRs influence host pathogen interactions in humans are unclear. The human-specificity of *S.* Typhi has meant reliance on infection of human cells in vitro or using non-typhoidal *Salmonella* in infections of mice. In human cells, toxin-induced DDRs arising from damage to nuclear and mitochondrial DNA causes cellular senescence leading to release of a host secretome referred to as the senescence-associated secretory phenotype (SASP) (Chen et al, 2024; ElGhazaly et al, 2023; Humphreys et al, 2020; Ibler et al, 2019). In mice, injection of purified toxin causes typhoid fever-like symptoms and fatality (Song et al, 2013) while infection with non-typhoidal *Salmonella* encoding typhoid toxin suppressed host damage and promoted chronic infections (Del Bel Belluz et al, 2016; Miller et al, 2018). To advance our understanding of typhoid toxin, typhoid fever was studied using a controlled human infection model, which involves deliberate infection of volunteers (Meiring et al, 2023). Human participants were challenged with either a wild-type (WT) *S.* Typhi strain expressing the toxin or a toxin-negative (TN) strain lacking the genes *pltB*, *pltA* and *cdtB* (Gibani et al, 2019). Counterintuitively, disease tended to be more severe in participants infected with *S.* Typhi-TN (severe typhoid in 7% due to *S.* Typhi-WT; 27% with *S.* Typhi-TN), which was reflected by significantly prolonged bacteraemia relative to participants infected with *S.* Typhi-WT (WT, 48 h; TN, 96 h).

The findings in Gibani et al, 2019 indicate that host responses to the typhoid toxin suppress the duration of bacteraemia but how toxin-induced DDRs might counteract infection is not known (Gibani et al, 2019). It is also not clear how host DDRs coordinate defences against pathogens themselves: attention has focused on mechanisms by which pathogens disarm the host DDR for microbial benefit (Siegl and Rudel, 2015), how bacterial genotoxins contribute to cancer (Lai et al, 2021) or trigger immune signalling pathways (Pons et al, 2021). It was thus hypothesised that host DDRs mount a defence against toxigenic pathogens such as *S.* Typhi, which could highlight how DDRs counteract bacterial pathogens. We sought to address whether toxin-mediated effects on the host proteome could explain the reduced duration of *S.* Typhi bacteraemia.

# Results

## Typhoid toxin manipulates the host secretome in bacteraemic humans with typhoid fever

Typhoid toxin induced a DDR-dependent host secretome in cultured fibroblasts, intestinal epithelial cells and macrophages (Chen et al, 2024; ElGhazaly et al, 2023; Ibler et al, 2019). Thus, it was hypothesised that in human participants challenged with WT toxigenic *S.* Typhi (Gibani et al, 2019), toxin-induced secretion would be reflected in the host proteome. Thus, we sought to identify proteomic signatures in response to typhoid toxin in samples harvested by Gibani and colleagues (Gibani et al, 2019). We performed LC-MS/MS analysis on plasma from bacteraemic participants at the time of typhoid diagnosis following infection with either WT (cyan) or TN (magenta) *S.* Typhi (Fig. 1A). As a reference for toxin-induced effects, proteomics was also performed on the same participants prior to infection (baseline), i.e., 20 participants before infection and 20 participants after infection. Plasma consists of high abundance proteins (~94%) conserved between individuals (Geyer et al, 2017), which were first removed

by immunodepletion to increase the dynamic range of the plasma proteome. LC-MS/MS identified 641 proteins at a 1% FDR (Dataset EV1), and label-free quantification was used to measure differences in the abundance of proteins between groups of participants (Datasets EV1 and EV2). After data filtering and normalisation, statistical analysis was performed on 440 proteins to identify significant differences between the groups using a permutation-based FDR of 0.05 (Dataset EV2).

In participants infected with TN *S.* Typhi, we found that 9 proteins were significantly different relative to baseline permutation-based FDR of 0.05 (Fig. 1B). In contrast, 41 host proteins were differentially regulated in response to the typhoid toxin during acute typhoid fever (Fig. 1C). Of the 9 TN-specific proteins, 6 proteins (CRP, LBP, LRG1, SAA1, TFRC) overlapped with the WT group identifying them as infection-specific and toxin-independent proteins (Fig. 1B, black text). Only 3 proteins (CST6, FSTL1, LYVE1) were unique to the TN group marking them as TN-specific (Fig. 1B,D, magenta). This contrasted with participants exposed to typhoid toxin as 35 proteins were enriched in the WT group (Fig. 1C,D, cyan), consistent with a toxin-mediated effect on the host secretome. This included 5 WT-specific proteins of increased abundance: beta-2-microglobulin (B2M), apolipoprotein C3 (APOC3), apolipoprotein F (APOF), lysozyme (LYZ), and thymosin (TMSB4X) (Fig. 1D), all of which are known secreted proteins (Uhlen et al, 2019). The remaining 30 proteins in the WT group were of decreased abundance (Fig. 1D). Taken together, the findings indicate that typhoid toxin manipulates the host proteome during acute typhoid fever.

## Typhoid toxin elicits secretion of APOC3 and LYZ during acute typhoid fever

We next investigated whether changes in protein abundance could indicate why *S.* Typhi bacteraemia was prolonged in the absence of the toxin. To narrow our focus, we concentrated on the 5 WT-specific proteins of increased abundance in acute typhoid fever (Fig. 1D: B2M, APOC3, APOF, LYZ, TMSB4X). B2M, APOF, and TMSB4X were most abundant but relative to *S.* Typhi-WT these proteins had also increased in response to *S.* Typhi-TN, albeit to a small extent with THSB4X. In contrast, APOC3 and LYZ increased in response to *S.* Typhi-WT in a toxin-dependent manner as both proteins were slightly reduced in response to *S.* Typhi-TN (Fig. 1D, see highlight). LYZ is a ubiquitous 15 kDa component of the innate immune response that hydrolyses β-1,4,glycosidic bonds in cell walls between *N*-acetylmuramic acid and *N*-acetylglucosamine in peptidoglycan causing bacterial lysis (Ragland and Criss, 2017). A role for Apolipoprotein C-III (APOC3) is less clear and required further investigation: APOC3 is a 9 kDa apolipoprotein only expressed in the liver within hepatocytes and epithelial cells of the gastrointestinal tract, which increases the concentration of free lipids in the blood (Norata et al, 2015). High concentrations of APOC3 are correlated with hypertriglyceridemia (Norata et al, 2015) but no role during bacterial infection is known.

## Toxin-induced DDRs mediate expression of APOC3 and LYZ

When we studied APOC3 in CACO2 intestinal cells, we found that very little APOC3 was observed in untreated control cells

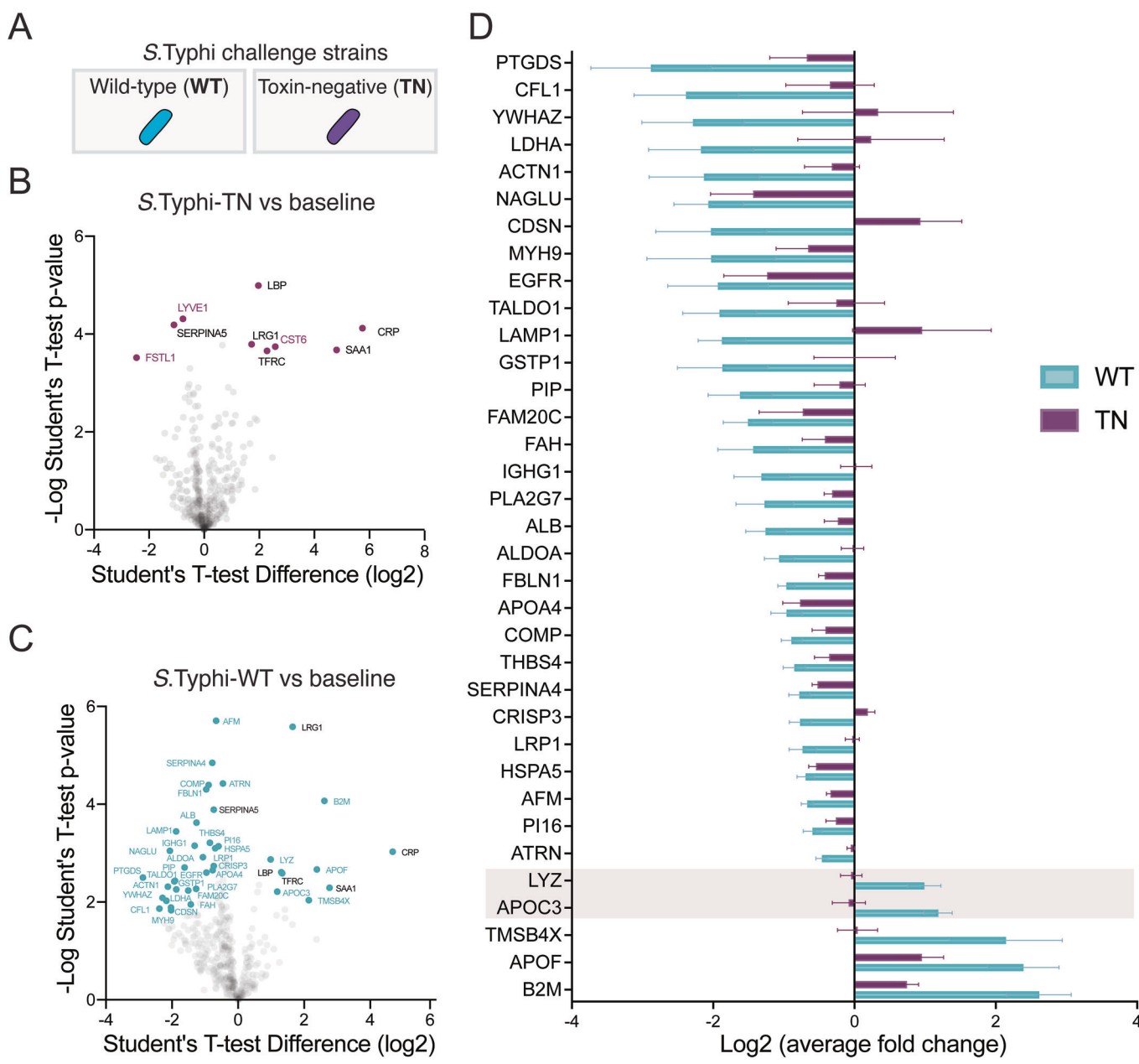

**Figure 1. Salmonella Typhi exhibits a toxin induced secretome in host organisms.**

(A) Schematic showing the strains used in human infection challenge study, S. Typhi-WT (cyan) expressing typhoid toxin and toxin-negative (TN; ΔpltA, ΔpltB, ΔcdtB) S. Typhi (purple). Volcano plots of plasma proteomics data showing human responses to infection by (B) S. Typhi TN, or (C) S. Typhi WT in bacteraemic participants at TD (time of diagnosis) relative to uninfected baseline. Toxin-dependent proteins identified (cyan text), toxin-independent proteins (purple), and proteins identified in both analyses (black) are indicated. Each point is the mean value from 10 biological replicates (participants). To identify proteins with a significant difference in protein abundance in infected participants compared to the uninfected baseline, the quantitative proteomic data were analysed using a unpaired two-sided Student's t test with a permutation-based FDR threshold of 0.05 to correct for multiple hypothesis testing. (D) Heatmap listing Log2-fold change of toxin-induced host proteins from (C) in participants infected with WT- (cyan) or TN- (purple) S. Typhi. LYZ and APOC3 highlighted (n = 10 participants).

(Fig. 2A,B). In contrast, CACO2 cells treated for 2 h with purified recombinant wild-type typhoid toxin (TxWT) expressed APOC3 at 96 h (Fig. 2A,B), which was observed in the nucleus as previously described (Soltysik et al, 2019). APOC3 expression was coincident with activation of DDRs marked by γH2AX (Fig. 2A,C), which corresponded to cell-cycle arrest as indicated by lack of DNA synthesis incorporating the nucleotide analogue EdU (Fig. 2A,D).

Indeed, APOC3 expression appeared dependent on DDRs as treating cells with typhoid toxin deficient in DNase activity due to its H160Q substitution (TxHQ) induced no γH2AX or APOC3 (Fig. 2A–C), which was consistent with EdU-positive nuclei marking replicating cells (Fig. 2A,D). Increased APOC3 expression in response to toxin-induced DDRs was mirrored by a relative increase in APOC3 secretion from the CACO2 cells at 96 h

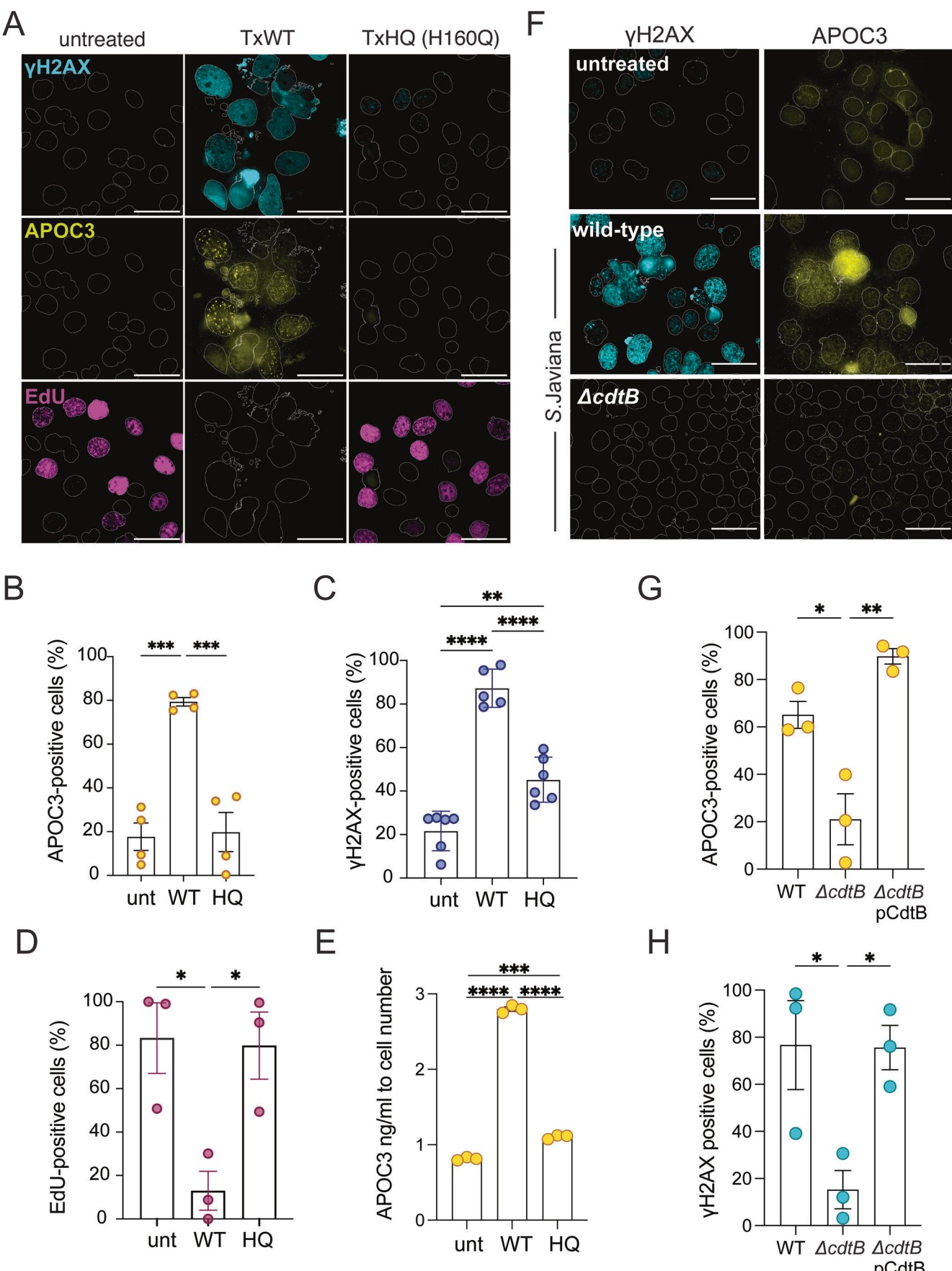

◄ **Figure 2.  Typhoid toxin-induced DNA damage and APOC3 secretion.**

(**A**) Representative images from three independent experiments of CACO2 intestinal cells either untreated, treated with wild-type typhoid toxin (TxWT) or H160Q DNase-deficient toxin (TxHQ) for 2 h prior to fluorescence microscopy at 96 h of γH2AX (cyan), APOC3 (yellow) and EdU (Magenta). EdU was incubated with cells 24 h before fixation. DAPI-stained nuclear outlines shown. Scale bars: 50 μm. (**B**) Bar chart showing proportion of APOC3 expressing cells ($n = 4$). (**C**) Bar chart showing proportion of γH2AX-positive cells ($n = 5$). (**D**) Bar chart showing proportion of cells incorporating EdU nucleotide analogue ($n = 3$). (**E**) ELISA of APOC3 secreted into growth media harvested from cells in (**A**) ($n = 3$). (**F**) Representative images from three independent experiments of CACO2 intestinal cells infected with wild-type or toxin-deficient (Δcdt B) S. Javiana for 1 h prior to incubation in gentamicin-containing media and imaging at 96 h. Immunofluorescence performed as (**A**). Scale bars: 50 μm. Bar charts showing (**G**) proportion of APOC3 expressing cells ($n = 3$), or (**H**) proportion of γH2AX-positive cells during infection from experiment in (**F**) ($n = 3$). Data also analysed following infection with Δcdt B expressing pTrc99A-cdtB (pCdtB). Statistical significance: one-way ANOVA with Tukey's test in (**B–E, G, H**) analysing all pairs of >3 groups. Data are presented as mean ± SEM. Asterisks indicate significance: *$P < 0.05$, **$P < 0.01$, ***$P < 0.001$, ****$P < 0.0001$. No significance (ns). Exact $P$ values in Appendix Table S1. Circles and n represent biological repeats. Source data are available online for this figure.

(Fig. 2E). In contrast, when we examined APOC3 inside intoxicated HepG2 liver cells, we found that cell-cycle arrest induced by TxWT had no effect on APOC3 that was expressed equivalently in all conditions (Fig. EV1A–C). The findings indicate that increased APOC3 originated from infected intestinal epithelial cells rather than liver cells during typhoid fever (Gibani et al, 2019). To test this possibility during infection, we examined APOC3 induction during infection with toxigenic *Salmonella* Javiana (Fig. 2F–H), a hazard group 2 non-typhoidal *Salmonella* serovar encoding typhoid toxin used for biosecurity reasons in place of the hazard group 3 pathogen *S.* Typhi. When CACO2 cells were infected with wild-type *S.* Javiana encoding typhoid toxin, APOC3 was observed in cells displaying increased levels of γH2AX relative to untreated and toxin-negative Δcdt B *S.* Javiana (Fig. 2F–H). We found that both γH2AX and APOC3 expression was restored during Δcdt B infection when the strain expressed cdtB from a plasmid demonstrating a dependency on CdtB (Fig. 2G,H). In summary, we find that toxin-induced DDRs modulates expression and secretion of APOC3, which was identified in human participants with acute typhoid fever.

We investigated whether APOC3 treatment of *S.* Javiana influenced pathogen growth but found no effect (Fig. EV1D). This suggests that APOC3 provides a marker of toxin-induced DDRs rather than playing a direct antimicrobial role against *Salmonella*. Consequently, we turned to LYZ, which has established antimicrobial activities through its ability to break down peptidoglycan in bacterial cell walls and through cationic pore-formation (Ragland and Criss, 2017).

We first determined whether LYZ was, like APOC3, also regulated by toxin-induced DDRs. Relative to untreated and TxHQ-treated control cells, we found that TxWT increased the proportion of LYZ-positive cells at 96 h (Fig. 3A,B), which indicated a role for toxin nuclease activity. The action of TxWT-induced nuclease activity was consistent with γH2AX-labelled DDRs and a lack of EdU incorporation into host cell DNA, showing cell cycle arrest. This contrasted with untreated controls that lacked γH2AX and synthesised EdU-positive DNA (Fig. 3A: untreated, TxHQ). In addition, LYZ was found at ~140 ng/ml in the media of TxHQ-treated cells, which was increased to ~280 ng/ml with TxWT (Fig. 3C). We reasoned that the LYZ in the secretome of TxHQ-treated cells may be due to FBS. Indeed, 10% FBS alone contained 150 ng/ml LYZ, which correlated with untreated (175 ng/ml) and the increase to 300 ng/ml by etoposide (Fig. EV2A), a topoisomerase inhibitor causing double-stranded DNA breaks and replication stress (Vesela et al, 2017). In line with these observations, we found that in 100% plasma of TYGER study participants (Gibani et al, 2019), the

concentration of LYZ increased from 27 μg/ml in non-infected participants to 40 μg/ml at the time of typhoid fever diagnosis due to wild-type *S.* Typhi (Fig. EV2B).

## LYZ causes *Salmonella* spheroplast formation and is augmented by lactoferrin

LYZ is best known for its ability to degrade peptidoglycan and induce cationic pore formation (Ragland and Criss, 2017). Hydrolysis of peptidoglycan and subsequent spheroplast formation are assisted by factors that mediate LYZ penetration into the periplasm of Gram-negative bacteria (Ragland and Criss, 2017). This includes factors in serum such as defensins and lactoferrin (LFN), which destabilise the bacterial cell wall facilitating LYZ entry (Chen et al, 2005; Ellison and Giehl, 1991; Panyutich et al, 1993; Ragland and Criss, 2017). LFN was found in all participants by proteomics suggesting the possibility that LYZ and LFN could work in combination to suppress *Salmonella* in response to typhoid toxin. Therefore, we examined whether LYZ and LFN generated spheroplasts. Transmission electron microscopy showed that untreated *S.* Javiana had a rod-shaped morphology, or a spherical morphology in ~50% of cases depending on bacterial cell orientation (Fig. 3D,E). When *S.* Javiana were treated with LFN alone, there was no significant difference and *Salmonella* morphology was equivalent to untreated while LYZ alone induced a small but significant increase in spheroplast formation (Fig. 3D,E). In contrast, LYZ and LFN in combination had a marked effect and increased the proportion of cells with spherical morphology to ~75%. This was equivalent in significance to the ~80% of spherical cells observed when LFN was replaced with EDTA that permeabilises the outer membrane of Gram-negative bacteria allowing entry of LYZ. In addition to inducing a round morphology typical of spheroplasts, LFN and LYZ caused instances of membrane damage where cell content lost interaction with its cell membrane resulting in protrusions (Fig. 3F, magenta arrows). Cell content was also released out of the cell resulting in the formation of a ghost-like shell (Fig. 3F, blue arrows).

We also examined spheroplast formation by fluorescence microscopy using *S.* Javiana expressing mCherry, which was challenging to observe due to the <5 μm size of bacteria (Fig. 3G, untreated). To observe spheroplasts more readily, we first treated *S.* Javiana with cephalexin, which inhibits cytokinesis causing an extended morphology (>5 μm) that is abolished by spheroplast formation due to degradation of peptidoglycan (Kawai et al, 2018; Renner, 2019; Sun et al, 2014) (Fig. 3G, cartoon left). In the cephalexin-treated control, elongated *S.* Javiana were observed by

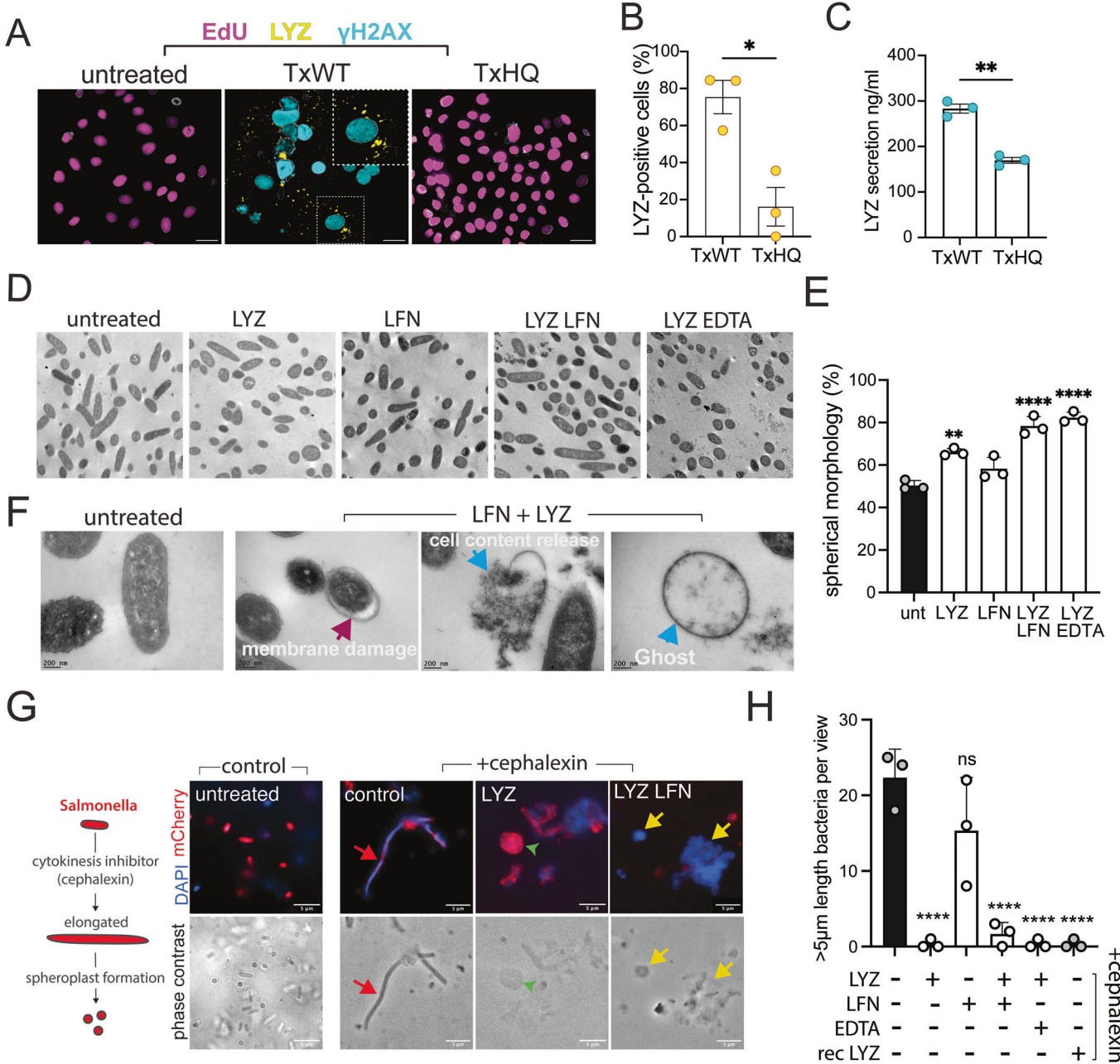

**Figure 3. Lysozyme secreted in response to typhoid nuclease activity and causes *Salmonella* spheroplast formation.**

(**A**) Representative fluorescence microscopy images of CACO2 cells either untreated or treated with 20 ng/ml TxWT or TxHQ for 2 h before imaging at 96 h. Images show γH2AX (cyan), LYZ (yellow) and EdU (magenta) with outlines of DAPI-stained nuclei. EdU nucleotide was incubated with cells for 24 h prior to fixation at 96 h. Magnified inset shows cell cycle-arrested cell producing LYZ. Scale bars: 50 μm. (**B**) Bar chart showing proportion of LYZ-expressing from cells in (**A**) (*n* = 3). (**C**) ELISA of LYZ secreted into growth media harvested from cells in (**A**) (*n* = 3). (**D**) Representative transmission electron microscopy (TEM) images from three independent experiments of *S.* Javiana either untreated or incubated with 1 mg/ml LYZ, 100 μg/ml LFN, LYZ and LFN, LYZ and 1 mM EDTA in M9 minimal media for 2 h. (**E**) Bar chart showing proportion of *S.* Javiana with spherical morphology from (**D**) (*n* = 3). (**F**) Representative TEM from (**D**) highlighting changes in cell morphology indicative of *S.* Javiana spheroplast formation with LYZ and LFN. (**G**) LYZ and LFN treatment of *S.* Javiana in the presence of cephalexin. Left: schematic of cell elongation due to cephalexin and spheroplast formation. Right: Representative fluorescence microscopy images, from three independent experiments, of *S.* Javiana pFPV-mCherry in LB at OD$_{600}$ 1.0 either untreated (left), or treated with cephalexin (right) prior to 20 min incubation with cephalexin only (control), LYZ, or LYZ and LFN. Top row: fluorescent images of DAPI-stained (blue) mCherry *S.* Javiana (red). Red arrows indicate elongated cephalexin-treated *S.* Javiana and yellow arrows spheroplasts incapable of mCherry retention due to LYZ and LFN. Bottom row: corresponding phase contrast images. Scale bars: 5 μm. Untreated, LYZ, and LYZ/LFN images reused in Fig. EV2C to show alongside additional controls. (**H**) Bar chart showing number of elongated *S.* Javiana (>5 μm) per field of view in (**G**) (*n* = 3). Statistical significance: Welch's unpaired *t* test in (**B, C**) for paired measures with unequal variances; one-way ANOVA with Dunnett's post hoc test in (**E**) for analysing >3 groups versus control, or with Brown–Forsythe test in (**H**) for unequal variances (>3 groups). Data are presented as mean ± SEM. Asterisks indicate significance *$P < 0.05$, **$P < 0.01$, ***$P < 0.001$, ****$P < 0.0001$. No significance (ns). Exact *P* values in Appendix Table S1. Circles and n represent biological repeats. Source data are available online for this figure.

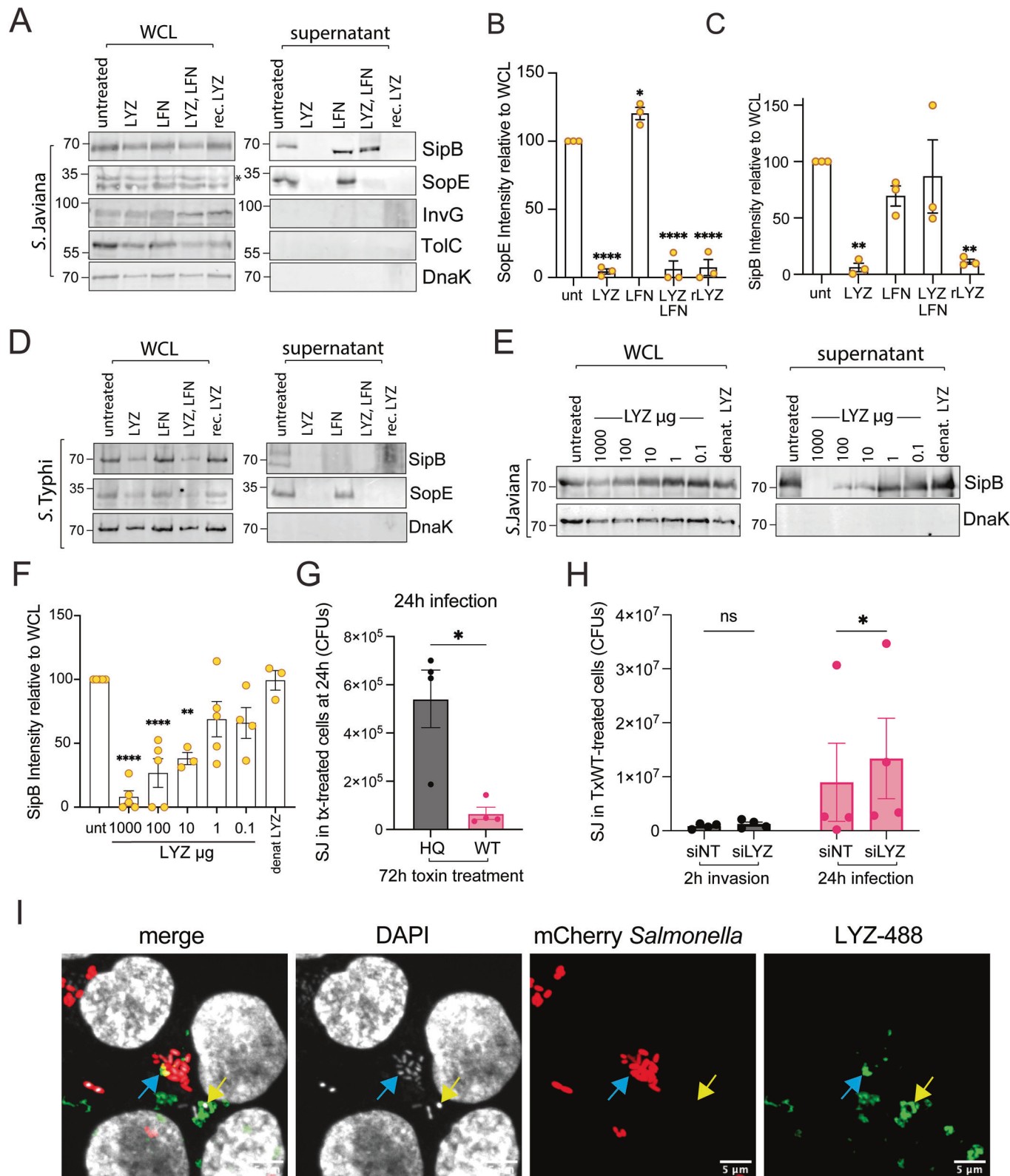

microscopy that contained mCherry and DAPI-stained DNA (Fig. 3G, red arrow). In the presence of LYZ and LFN however, the elongated morphology was absent (Fig. 3G, bottom). Instead, round *Salmonella* that failed to retain mCherry were found by

DAPI staining (yellow arrow), which demonstrated spheroplast formation and is consistent with cell content release observed by electron microscopy (Fig. 3F: LFN, LYZ). The number of elongated *Salmonella* was quantified, which confirmed that LYZ and LFN

**Figure 4. Host antimicrobial responses triggered by typhoid toxin.**

(A) Representative immunoblot from three independent experiments of *S.* Javiana in LB broth either untreated, or cultured with 1 mg/ml endogenous LYZ, 100 µg/ml LFN, LYZ and LFN, or 1 mg/ml recombinant LYZ (rec. LYZ) for 2 h. Whole cell lysates (WCLs) or supernatants immunoblotted with antibodies to virulence effectors SipB or SopE, T3SS component InvG, T1SS component TolC or the intracellular loading control DnaK. Molecular weight (MW) markers left. *indicates unidentified cross-reactive protein in SopE blot of whole cell lysate. Quantification of immunoblot band intensity quantified from $n = 3$ of experiment (A) for: (B) SopE, or (C) SipB. (D) Same experiment as (A) using attenuated *S.* Typhi BRD948, $n = 2$. (E) Immunoblot of whole cell lysates or supernatants of three independent experiments from *S.* Javiana cultured in LB only (untreated) or treated for 2 h with indicated concentrations of LYZ, or denatured LYZ (denat. LYZ). Immunoblotted with antibodies to SipB or DnaK. MW in kDa, left. (F) Quantification of immunoblot band intensity from $n = 3$ of experiment in (E). (G) *S.* Javiana (SJ) CFUs calculated on LB agar plates at 24 h post-infection of CACO2 cells already treated for 72 h with TxWT or TxHQ ($n = 4$). (H) *Salmonella* infection of LYZ-depleted intoxicated cells. HCT116 cells were transfected with non-targeting or LYZ siRNA (siNT; siLYZ) for 48 h before treatment with TxWT and further 48 h incubation (96 h total). At 96 h, cells were infected with *S.* Javiana (SJ) and CFUs quantified at 2 h or 24 h on LB agar plates ($n = 4$). (I) Localisation of endocytosed LYZ during *S.* Javiana infection ($n = 1$). HCT116 cells were infected with *S.* Javiana pFPV-mCherry (mCherry *Salmonella*) for 30 min (MOI 100) when 100 µg/ml LYZ-488 was added to infected cells to allow endocytosis and incubated for 2 h in gentamicin-containing media ($n = 3$). Outlines of DAPI-stained nuclei. Arrows indicate colocalisation. Scale bar: 5 µm. Statistical significance: one-way ANOVA with Dunnett's post hoc test (B, C, F) for analysing >3 groups versus control; Welch's unpaired *t* test (G) for paired measures with unequal variances; two-way ANOVA Sidak's multiple comparison in (H) assessing two independent variables. Data are presented as mean ± SEM. Asterisks indicate significance: *$P < 0.05$, **$P < 0.01$, ***$P < 0.001$, ****$P < 0.0001$. No significance (ns). Exact *P* values in Appendix Table S1. Circles and n represent biological repeats. Source data are available online for this figure.

reduced the number of elongated *Salmonella* (Fig. 3H). The same trend was observed when LFN was replaced with EDTA. In contrast, LFN alone had no effect as elongated *Salmonella* were still observed (Figs. 3H and EV2C). We also found that LYZ alone caused a loss in cell morphology in *S.* Javiana (Fig. 3G,H), which retained mCherry, indicating no membrane damage, and contrasted with LYZ and LFN together. This agrees with observations by electron microscopy where LYZ increased the proportion of spherical cells (Fig. 3D,E) and suggests limited entry of LYZ into the periplasm. The same trends were observed in *S.* Typhi and *S.* Typhimurium where LYZ induced spheroplast formation alone or in combination with LFN or EDTA (Fig. EV2D,E).

## LYZ suppresses the function of the *Salmonella* type 3 secretion system

*Salmonella* infects cells using a T3SS, which is structured across the inner and outer membranes of the Gram-negative bacterium for injection of virulence effectors into host cells (Kubori et al, 1998). Thus, we asked whether LYZ and LFN-induced changes in morphology influence T3SS-mediated secretion of virulence effectors SipB and SopE that play key roles in invasion (McGhie et al, 2009). LYZ and LFN were added to the cultures of toxigenic *S.* Javiana and attenuated *S.* Typhi before analysing the expression and secretion of SipB and SopE (Fig. 4A–D). In the untreated control, we found that both *S.* Javiana and attenuated *S.* Typhi expressed SipB and SopE (Fig. 4A,D). The untreated culture supernatant contained secreted SipB and SopE, which, as expected, lacked the intracellular loading control DnaK that was present in the whole cell lysate (WCL). LYZ and LFN in combination did not impair secretion of SipB in *S.* Javiana but did significantly inhibit SopE (Fig. 4A–C). The effects of LYZ were mirrored in *S.* Typhimurium where SopE secretion was impeded (Fig. EV3A). The effect of LYZ was more striking in attenuated *S.* Typhi where secretion of both SipB and SopE was inhibited by LYZ and LFN (Fig. 4D). *S.* Typhi is surrounded by the Vi capsule that protects against bacterial cell lysis in serum (Looney and Steigbigel, 1986). However, the Vi capsule provided no protection against LYZ as SipB secretion was inhibited in both Vi-expressing and VI-deficient *S.* Typhi (Fig. EV3B,C). Thus, spheroplast formation impairs the function of the T3SS.

In control experiments, we found that LFN alone had no effect on T3SS-mediated secretion in *Salmonella* as secreted SipB and

SopE were detected (Figs. 4A,D and EV3A). To our surprise however, we found that LYZ alone impaired T3SS as SipB and SopE secretion was lost in *S.* Javiana, *S.* Typhi and *S.* Typhimurium (Figs. 4A–D and EV3A). The same trend was observed when recombinant LYZ (rec.LYZ) replaced the endogenous conventional lysozyme (i.e., LYZ). Extracellular LYZ concentrations range from 10 µg/ml in the serum of healthy adults to 1200 µg/ml in tears (Hankiewicz and Swierczek, 1974). Thus, we examined secretion of SipB by *S.* Javiana treated with indicated concentrations of LYZ (Fig. 4E,F). We found that 1000, 100 and 10 µg/ml LYZ significantly inhibited secretion of SipB but not 1 or 0.1 µg/ml LYZ, or denatured LYZ, that were equivalent to the untreated control. In *S.* Javiana, the inhibitory effect on SipB secretion was significant with LYZ alone but not LYZ and LFN in combination (Fig. 4A,C).

It is possible that LYZ-induced cationic pore formation caused loss of secretion via the T3SS rather than spheroplast formation. However, we observed that LYZ and rec.LYZ each induced spheroplast formation indicating penetration of lysozyme into the periplasm of *Salmonella* (Fig. 3H). We also found no evidence of membrane damage as mCherry was retained by LYZ-treated *S.* Javiana but was lost following LYZ/LFN-treatment (Fig. 3G). Moreover, *S.* Javiana outer membrane proteins InvG and TolC were retained in the WCL rather than liberated into the supernatant (Fig. 4A). We sought to examine the effect of LYZ on secretion of the T1SS substrate SiiE (Gerlach et al, 2007) but immunoblotting experiments were unsuccessful. Instead, we reasoned that the inhibitory effect of LYZ might extend to assembly of flagella, which is driven by a distinct T3SS that exports flagella rather than secretes virulence effectors (Diepold and Armitage, 2015) However, when flagella were broken mechanically by shear forces, we found that LYZ had no effect on the export of flagellin (Fig. EV3D), which indicates that LYZ activity mediates a specific effect on virulence effector secretion. Taken together, the results indicate that toxin-induced secretion of LYZ impairs virulence mechanisms of *Salmonella* alone or in combination with LFN.

## Host responses to typhoid toxin mediate an intracellular antimicrobial defence

Our findings so far indicate that toxin-induced DDRs cause a host-mediated antimicrobial response, which is signified by secretion of LYZ. To test this during *Salmonella* infection, we first treated

intestinal cells with TxWT for 72 h to trigger an antimicrobial response prior to a 24 h infection with *S.* Javiana (Fig. 4G). Relative to TxHQ-treated cells, we found that TxWT treatment significantly reduced *Salmonella* infection (Fig. 4G: $6 \times 10^5$ CFUs in TxHQ; $1 \times 10^5$ CFUs in TxWT) showing an antimicrobial response. The same trend was observed in HCT116 cells (Fig. EV3E). To determine whether LYZ contributes to antimicrobial defences, we depleted LYZ by transfecting HCT116 cells with non-targeting (siNT) or LYZ (siLYZ) siRNA before treatment with TxWT (Fig. EV3F). We found that LYZ had no effect on *Salmonella* invasion into LYZ-depleted host cells (Fig. 4H). On reflection, this was not unexpected as TxWT-treated cells secreted ~280 ng/ml LYZ (Fig. 3C), which was in the range of LYZ concentrations that did not significantly inhibit the *Salmonella* T3SS (Fig. 4H). Nevertheless, the majority of *Salmonella* are intracellular in the bloodstream (Wain et al, 1998), which is a phase of infection governed by virulence effectors delivered by T3SSs encoded by SPI-1 and SPI-2 (McGhie et al, 2009). When we examined *Salmonella* infected TxWT-treated cells at 24 h, LYZ-depletion significantly increased infection relative siNT control indicating that LYZ inhibits intracellular *Salmonella* infections (Fig. 4H: $8 \times 10^6$ CFUs in siNT; $1.4 \times 10^7$ CFUs in siLYZ). Consistent with this, we observed that exogenous LYZ-488 was endocytosed and localised to intracellular *S.* Javiana-mCherry during infection (Fig. 4I, blue arrows). We also observed LYZ localised to *Salmonella* that had lost mCherry (Fig. 4I, yellow arrows), which was previously observed during spheroplast formation (Fig. 3F,G). In summary, the results show that toxin nuclease activity elicits a host antimicrobial defence that counteracts intracellular *Salmonella* infection with significant inhibition mediated by LYZ.

## Typhoid toxin triggers LYZ expression in diverse cell types

Our findings so far indicate that toxin nuclease activity triggered expression of APOC3 and LYZ in CACO2 intestinal cells. Thus, we investigated the signalling cascade in more detail and in divergent cell types. To counteract pathology, the DDR is activated through kinases ATM (ataxia-telangiectasia mutated), which responds to DSBs, and ATR (ATM and rad3-related) that senses single-strand DNA breaks (Polo and Jackson, 2011). Both ATM/ATR are inhibited by caffeine (Sarkaria et al, 1999). We used caffeine to inhibit ATM/ATR and assess their role in APOC3 and LYZ expression in CACO2 cells treated with either TxWT or ETP at 48 h (Fig. 5A,B). In contrast to untreated control cells, we found that either TxWT or ETP induced γH2AX-labelled DDRs (Fig. 5A), which were associated with expression of APOC3 and LYZ divergently distributed inside the damaged cells (Fig. 5B–D). Uniting the functions of ATM and ATR is phosphorylation of their effector γH2AX (Polo and Jackson, 2011), which was used as a control for DDRs. In the presence of caffeine, the γH2AX response to either TxWT or ETP was suppressed indicating inhibition of ATM/ATR (Figs. 5A and EV4A). Caffeine treatment also disabled the ability of toxin-induced DDRs to drive expression of APOC3 and LYZ, a phenotype also observed with ETP (Fig. 5B–D), which indicates a role for ATM/ATR-mediated DDRs.

We investigated toxin-mediated LYZ expression in diverse intestinal cell lines at 72 h (Fig. 5E), namely CACO2, HCT116 and RKO. Relative to TxHQ, TxWT induced γH2AX labelled DDRs in

CACO2 and HCT116 cells, which corresponded with increased LYZ expression. We found that RKO cells were less sensitive to TxWT and exhibited only modest γH2AX signalling, which was consistent with reduced LYZ expression. This was not the case for ETP, which increased γH2AX and LYZ in each cell line relative to the untreated control. These findings demonstrate that typhoid toxin nuclease activity causes increased LYZ expression in diverse intestinal cell lines.

We also examined liver epithelial cells and macrophages (Fig EV4), which contribute to the protein content of plasma and are sites of disseminated *Salmonella* infections. We found that HepG2 liver epithelial cells behaved much like intestinal cells as toxin- or ETP-induced γH2AX-labelled DDRs coincided with increased LYZ expression (Fig. EV4B). We next investigated macrophages. Previously, we reported that the toxin caused single-strand DNA breaks during DNA synthesis resulting in replication stress, which was observed in precursor replicating THP1 monocytes (Mo) but not differentiated non-replicating THP1 macrophage (MΦ) cells (Ibler et al, 2019). Once again, we observed no γH2AX-labelled DDRs in response to TxWT in non-replicating MΦ cells, which was equivalent to TxHQ (Fig. EV4C). In contrast, TxWT, but not TxHQ, caused DDRs in replication competent Mo cells, which coincided with increased LYZ expression. This agrees with our observations in epithelial cells where LYZ expression was increased by ETP and typhoid toxin (Figs. 5E and EV4B), both of which cause replication stress (Ibler et al, 2019; Vesela et al, 2017). These results show that DNA damage induced by typhoid toxin increases LYZ expression in liver and intestinal epithelial cells, and monocytes, which can be triggered by replication stress.

## Toxin-mediated p53 activation regulates expression and secretion of LYZ

We previously showed that typhoid toxin induced cellular senescence marked by expression of p21 and secretion of the SASP protein GDF15 (ElGhazaly et al, 2023). GDF15 and p21 are both senescence effectors of p53, which suppresses tumour development following DNA damage via apoptosis and senescence (Kumari and Jat, 2021; Siegl and Rudel, 2015). We found that increased LYZ expression coincided with elevated p21 and GDF15 in HCT116, RKO and HepG2 cells, which exhibited elevated p53 (Fig. 5E and EV4B). In p53-deficient CACO2 and THP1 cells, p53 effectors p21 and GDF15 were absent but not LYZ (Figs. 5E and EV4E), which indicates that LYZ is not an effector of p53 but may be regulated by p53 in a cell type-dependent manner. Sure enough, LYZ was expressed and secreted in wild-type HCT116 cells treated with TxWT at 72 h, which was significantly inhibited in TP53-knockout HCT116 cells (Fig. 6A–C). The findings were supported by immunoblotting where the absence of p53 in TxWT-treated TP53-knockout HCT116 cells corresponded with loss of LYZ, p21 and GDF15 (Fig. 6D). Loss of the p53 signalling cascade was not due to lack of DNA damage since γH2AX was elevated in a toxin-dependent manner and had increased in TP53-knockout HCT116 cells. In summary, LYZ is triggered as part of p53-dependent senescence innate immune responses in HCT116 intestinal cells.

## The host cell suppresses toxin-mediated oxidative stress via LYZ

To provide more mechanistic insight, we investigated the connection between LYZ and DDRs. DNA damage due to typhoid

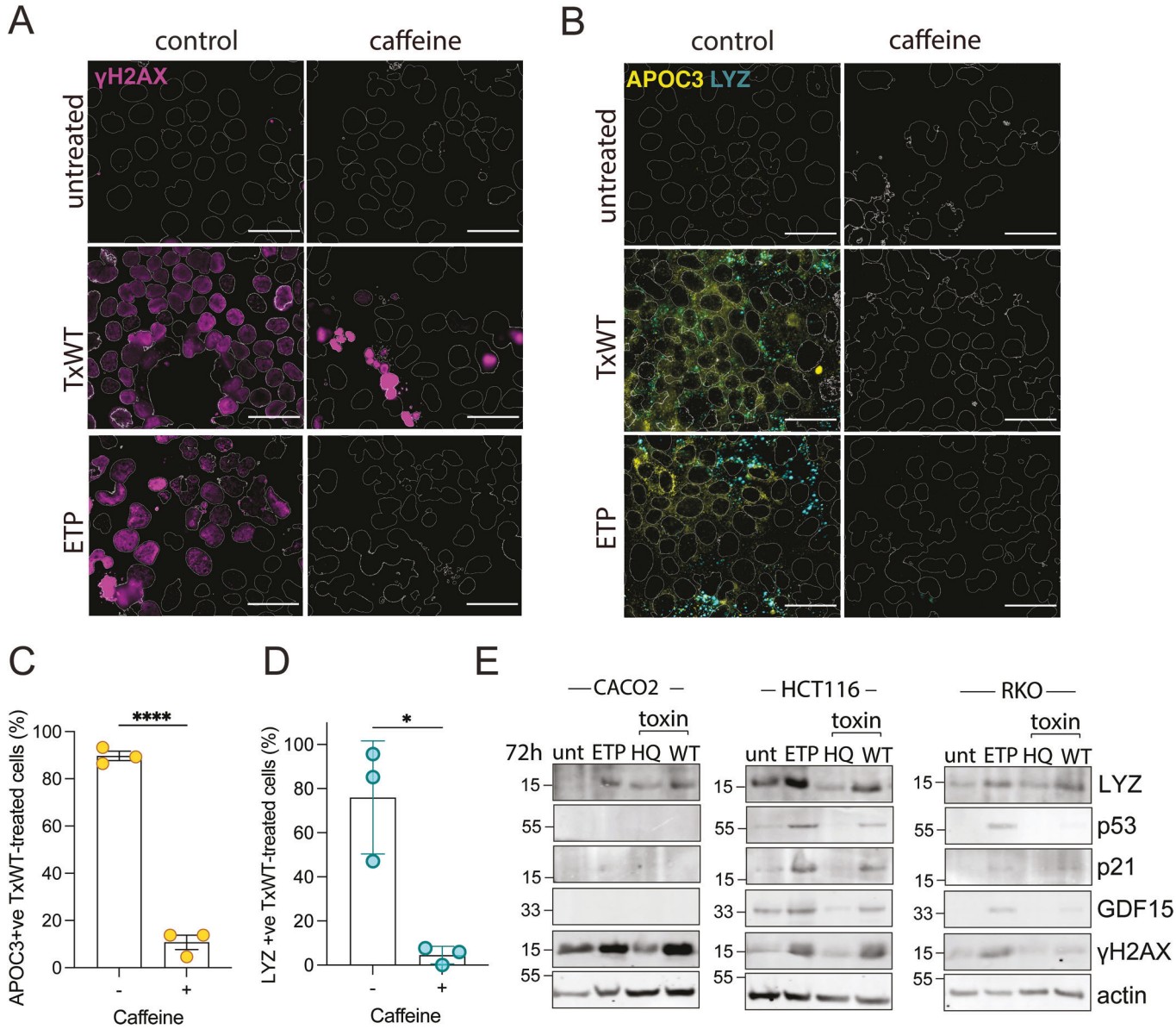

**Figure 5. Toxin-induced DNA damage responses mediate APOC3 and LYZ expression.**

(A) Fluorescence microscopy images of CACO2 cells treated with fresh media (untreated), 20 ng/ml TxWT or 8 μM etoposide (ETP) for 2 h, −/+ caffeine, at 48 h. Representative images, from three independent experiments, show γH2AX (magenta) and outlines of DAPI-stained nuclei. Scale bars: 50 μm. (B) Same experiment as (A) with imaging of APOC3 (yellow) and LYZ (cyan) (n = 3). (C) Bar chart showing proportion of APOC3-positive cells or (D) LYZ-positive puncta per field of view, from experiment in (B) (n = 3). (E) LYZ expression in CACO2, HCT116 and RKO intestinal epithelial cells at 72 h following no treatment (unt), or treatment with etoposide, TxHQ or TxWT (n = 2). Immunoblots performed with indicated antibodies. MW in kDa, left. Statistical significance: Welch's unpaired t test for paired measures with unequal variances in (C, D). Data presented as mean ± SEM. Asterisks indicate significance: *P < 0.05, **P < 0.01, ***P < 0.001, ****P < 0.0001. Exact P values in Appendix Table S1. Circles and n represent biological replicates. Source data are available online for this figure.

toxin activates the type-1 interferon (IFN) response (Chen et al, 2024), and IFN can activate p53 expression (Takaoka et al, 2003). However, control experiments showed that typhoid toxin did not induce IFN responses in HCT116 cells (Fig. EV4D), which means LYZ and p53 expression were IFN-independent.

Typhoid toxin was recently shown to cause mitochondrial injury resulting in mitochondrial oxidative stress and senescence (Chen et al, 2024). Mitochondrial oxidative phosphorylation drives ATP production

but its elevation due to nuclear DNA damage produces reactive oxygen species (ROS) (Fang et al, 2016; Xu et al, 2025), a phenomenon counteracted by p53-induced antioxidants (Liu and Gu, 2022; Sablina et al, 2005). Thus, we inhibited mitochondrial oxidative phosphorylation to impede ROS production in TxWT-treated cells at 24 h using the drug CCCP (Carbonyl cyanide 3-chlorophenylhydrazone) before examining p53 responses and LYZ (Fig. 6E–G: CCCP). We found that CCCP inhibited TxWT-induced production of LYZ, p53 and downstream

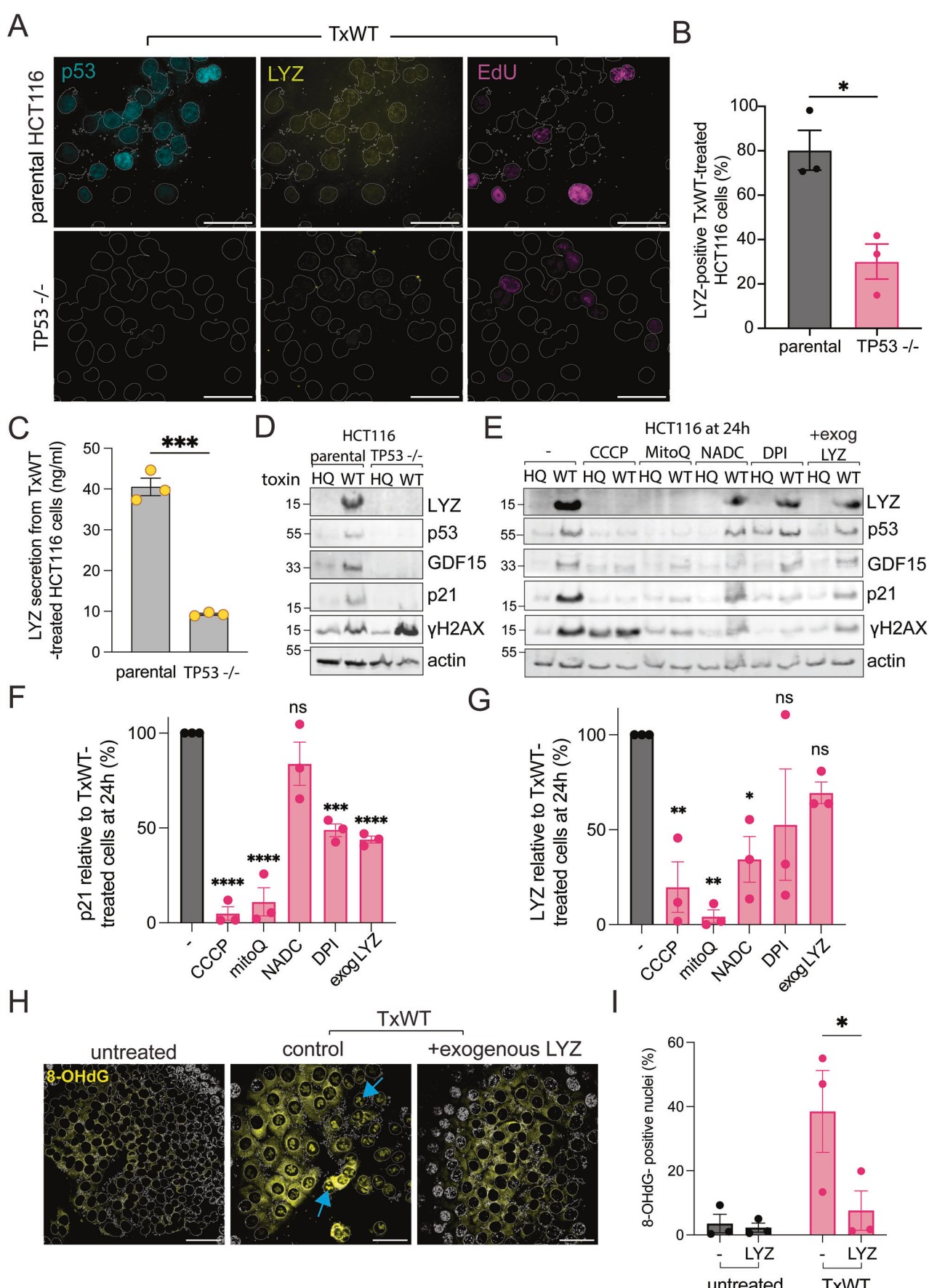

**Figure 6. LYZ regulation by p53 and its influence on oxidative stress.**

(A) Fluorescence microscopy, from there independent experiments, of parental and TP53 -/- HCT116 cells treated with TxWT prior to imaging at 72 h of p53 (cyan), LYZ (yellow) and EdU (magenta). EdU was incubated with cells 24 h before fixation. DAPI-stained nuclear outlines shown. Scale bars: 50 μm. Bar chart quantifying (B) LYZ-positive cells ($n = 3$), or (C) LYZ secretion from cells ($n = 3$), from experiment (A). (D) Immunoblot showing LYZ expression and p53 responses in parental and TP53 $-/-$ HCT116 cells at 72 h following 2 h treatment with TxHQ or TxWT ($n = 2$). Antibodies indicated. MW markers left. (E) Immunoblot showing LYZ expression and p53 responses in HCT116 cells at 24 h following treatment with TxHQ or TxWT and incubation in complete media ($-$) or with addition of oxidative stress inhibitors CCCP, MitoQ, NADC, or DPI ($n = 3$). Antibodies indicated. MW markers left. Bar chart quantifying (F) p21 or (G) LYZ, band intensities from $n = 3$ of the experiment in (E). (H) Fluorescence microscopy images of oxidative stress in TxWT-treated HCT116 cells from three independent experiments, alone (control), or in the presence of added 1 μg/ml LYZ (+ exogenous LYZ). Oxidative damage to DNA determined using antibodies to 8-OHdG (yellow). Untreated control HCT116 cells and outlines of DAPI-stained nuclei shown. Arrows indicate nuclear 8-OHdG. Scale bars: 50 μm. (I) Bar chart quantifying 8-OHdG-positive nuclei from cells in experiment (H) ($n = 3$). Statistical significance: Welch's unpaired $t$ test for paired measures with unequal variances in (B, C); one-way ANOVA Dunnett's post hoc test (F, G, I) for analysing >3 groups versus control. Data are presented as mean ± SEM. Asterisks indicate significance: $*P < 0.05$, $**P < 0.01$, $***P < 0.001$, $****P < 0.0001$. No significance (ns). Exact $P$ values in Appendix Table S1. Circles and n represent biological replicates. Source data are available online for this figure.

senescence p21 and GDF15 responses. However, nuclear γH2AX-labelled DDRs were still observed in the presence of CCCP, even in TxHQ controls, suggesting that inhibiting oxidative phosphorylation caused DDRs independently of toxin nuclease activity. This complicated interpretation and thus we investigated oxidative stress using alternative inhibitors.

We examined oxidative stress in mitochondria further using mitoquinone mesylate (MitoQ), which scavenges mitochondrial ROS thereby inhibiting oxidative stress (Dhanasekaran et al, 2004). Once again, we observed that inhibiting mitochondrial ROS, this time via MitoQ, markedly suppressed p53 expression and resulting induction of p21, GDF15 and LYZ in response to TxWT (Fig. 6E–G, MitoQ). Interestingly, this time we found that inhibiting mitochondrial ROS impeded TxWT-induced γH2AX suggesting ROS originating from mitochondria was required for nuclear DDRs driving LYZ expression. Thus, we explored oxidative stress further using N-acetyl-D-Cysteine (NADC), which scavenges ROS via its thiol group (Jones et al, 1995), and diphenyleneiodonium (DPI), an NADPH oxidase inhibitor that also inhibits mitochondrial ROS production (Li and Trush, 1998). NADC and DPI inhibited TxWT-induced γH2AX and the p53-p21-GDF15 axis (Fig. 6E–G, NADC), although the effects were modest relative to MitoQ as TxWT-dependent DDRs relative to TxHQ were apparent. As a result of the remaining TxWT-dependent DDRs in NADC- or DPI-treated cells, LYZ expression was still observed. Taken together, the findings indicate that mitochondrial ROS were required for toxin-mediated DDRs that trigger p53-dependent senescence responses and LYZ production.

## LYZ suppresses nuclear oxidative damage triggered by typhoid toxin

Our data indicate that host sensing of toxin-induced DNA damage triggers an antimicrobial response marked by LYZ that counteracted intracellular *Salmonella* infection. We also found that LYZ expression could be promoted by p53. Many bacterial pathogens deactivate p53 (Siegl and Rudel, 2015), but, nevertheless, it was not immediately clear why the tumour suppressor p53 influences expression of a dedicated antimicrobial. Interestingly, LYZ has been shown to have free radical scavenging activity and can reduce oxidative stress (Chen et al, 2025; Liu et al, 2006). Moreover, ROS can be transmitted between neighbouring cells (Fichman et al, 2023), which would indicate a role for extracellular ROS scavengers.

To investigate whether LYZ can suppress toxin-induced oxidative stress, we treated HCT116 cells with control TxHQ or TxWT for 24 h

with or without addition of exogenous LYZ. We found that addition of LYZ significantly suppressed toxin-induced senescence responses, which was exemplified by reduction in p21 and γH2AX DDRs (Fig. 6E,F, exog LYZ). To investigate whether the inhibitory effect was due to reduced oxidative stress, we examined the same experiment by fluorescence microscopy after labelling with antibodies to 8-Hydroxydeoxyguanosine (8-OHdG) (Fig. 6H,I), which marks oxidation of guanine that is a DNA lesion arising from oxidative stress (Wu et al, 2004). 8-OHdG was observed in mitochondria surrounding cell nuclei in all conditions but in TxWT-treated control cells, 8-OHdG was also found in inside 40% of nuclei (Fig. 6H,I, blue arrows). Nuclear 8-OHdG in TxWT-treated cells was significantly reduced to ~10% when cells were incubated with LYZ (Fig. 6H,I), which was consistent with immunoblotting experiments showing that LYZ dampens TxWT-induced nuclear responses (Fig. 6E,F, exog LYZ). The results indicate that LYZ counteracts toxin-induced senescence responses by inhibiting oxidative stress in the nucleus, which suggests DDRs elicit antimicrobial responses that suppress pathogen-induced damage.

## Discussion

Human infection challenge studies comparing wild-type and typhoid toxin-negative *S*. Typhi showed that participants infected with wild-type *S*. Typhi had significantly shorter duration of bacteraemia (Gibani et al, 2019), a hallmark of typhoid fever (Meiring et al, 2023). Thus, we hypothesised that toxin-induced DDRs may induce an innate immune response that suppresses infection. Our study demonstrates that participants infected with wild-type *S*. Typhi induce a divergent host secretome, which contains effectors of innate immunity including the antimicrobial lysozyme (Fig. 7: Proposed model). Indeed, we observed that toxin-mediated DDRs triggered secretion of lysozyme in cultured cells. Lysozyme inhibited the function of the *Salmonella* T3SS and intracellular infections in a toxin-dependent manner. Lysozyme expression was stimulated by oxidative stress, regulated by p53, and could counter ROS-mediated oxidation of DNA bases. Thus, lysozyme can counteract *Salmonella* in two ways: (i) inhibiting intracellular infection, and (ii) dampening toxin-induced oxidative stress and resulting DDRs. This is the first time typhoid toxin has been associated with eliciting host defence mechanisms that protect humans from typhoid fever.

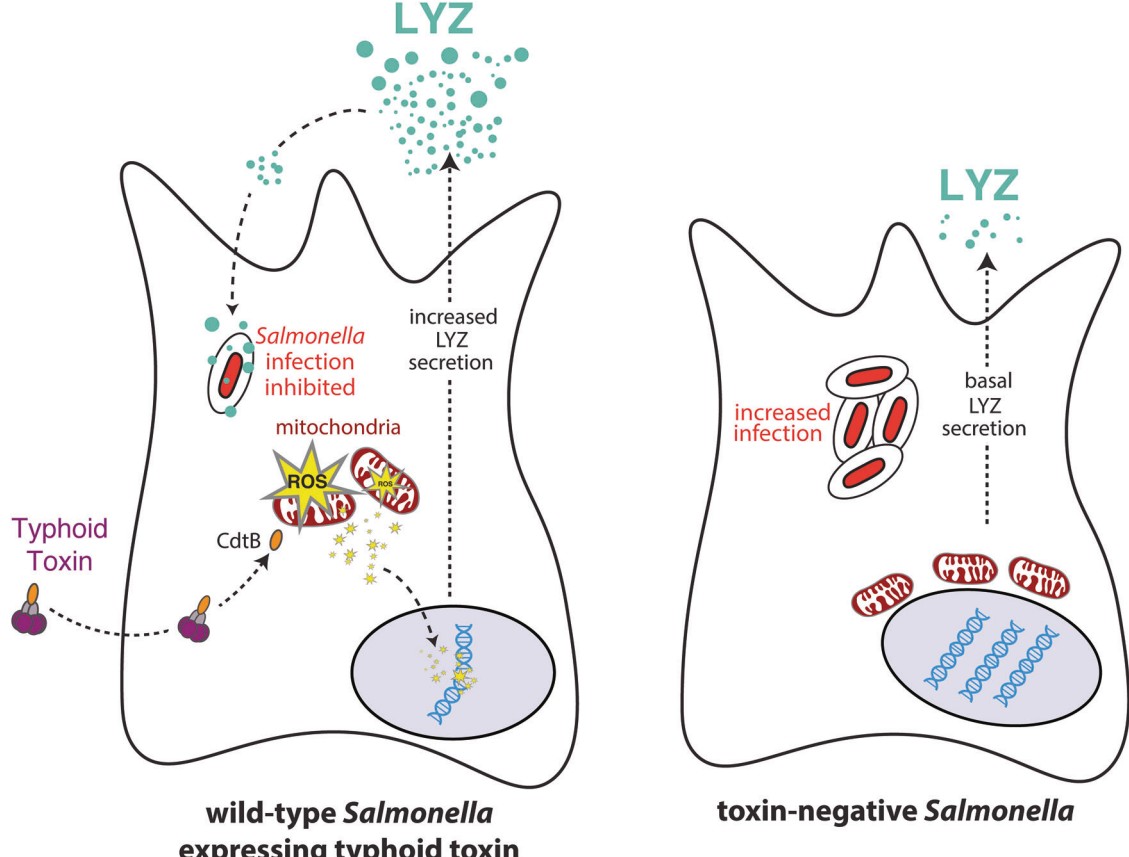

**Figure 7. Toxin-induced damage to the host cell triggers antimicrobial defence.**

Proposed model showing uptake of typhoid toxin during infection by wild-type *Salmonella* (left) relative to toxin-negative *Salmonella* infection (right). The nuclease activity of CdtB causes mitochondrial ROS that leads to nuclear DNA damage responses. The resulting increased expression and secretion of lysozyme (LYZ) inhibits intracellular infections.

The findings from this study support the view that the activity of typhoid toxin comes at a cost for *Salmonella* during typhoid fever, which leaves the role of the typhoid toxin unclear. Indeed, *S.* Paratyphi B does not encode typhoid toxin but can cause a typhoid-like disease (den Bakker et al, 2011), which was observed with *S.* Typhi-TN (Gibani et al, 2019). It is possible that the presence of typhoid toxin in the blood causes fatalities, which has been observed in mouse models (Fowler et al, 2019; Lee et al, 2020; Song et al, 2013), and is a symptom associated with severe typhoid fever in humans (Meiring et al, 2023). However, human infection challenge studies cannot test severe typhoid fever for ethical reasons (Gibani et al, 2019). It is also possible that the cost of activating host defences is balanced somewhat by the role of typhoid toxin in persistent infections and systemic spread, which has been observed during mouse infection models (Del Bel Belluz et al, 2016; Miller et al, 2018). Indeed, this study further highlights the arms race between host and pathogen by providing an example of how infection is counteracted during typhoid fever.

Lysozyme expression was mediated through DDRs in all cell types but was controlled via p53-dependent and p53-independent

pathways in a cell type-specific manner. There are likely multiple causes of DNA damage that lead to lysozyme expression: conditions associated with replication stress triggered LYZ expression in THP1 monocytes and etoposide-treated epithelial cells, while we observed that oxidative stress triggered LYZ expression in HCT116 intestinal epithelial cells. Typhoid toxin has recently been shown to cause oxidative stress via mitochondrial ROS resulting in senescence (Chen et al, 2024). However, damage to nuclear DNA can also cause mitochondrial ROS, and vice versa (Fang et al, 2016; Xu et al, 2025). When we tested this, we found that mitochondrial ROS was required for nuclear DNA damage, which was exemplified by findings using MitoQ and builds upon the observations by Chen and colleagues (Chen et al, 2024).

Oxidative stress explains why p53 signalling was activated in response to typhoid toxin but not necessarily lysozyme, which is regarded as an antimicrobial rather than tumour suppressor. However, p53 mediates expression of transcription factors and antioxidants that counteract oxidative stress (Liu and Gu, 2022; Sablina et al, 2005), and our findings indicate that lysozyme can inhibit oxidative stress. This is consistent with evidence that lysozyme can scavenge free radicals (Chen et al, 2025; Liu et al,

2006), and may play a role as a SASP protein neutralising extracellular ROS, a damage-associated molecular pattern (DAMP) that damages adjacent cells (Fichman et al, 2023). Though SASP from senescent cells has not been investigated for its antimicrobial properties, we propose that lysozyme fits within the diverse type of secreted antimicrobial factors already identified as SASP factors (e.g. IFNγ, IFN-1, TNFα, IL-1β, and CCN1 (Boxx and Cheng, 2016; Ingram et al, 2017; Wemyss and Pearson, 2019; Zganiacz et al, 2004). Indeed, we found that lysozyme was expressed via p53-dependent DDRs alongside senescence effectors p21 and GDF15; the latter being a previously identified SASP protein secreted by toxin-treated cells (ElGhazaly et al, 2023).

Lysozyme and lactoferrin together caused membrane damage that inhibited the T3SS. In contrast, lysozyme alone specifically inhibited virulence T3SSs rather than via extensive membrane damage, which could be mediated via an unknown mechanism or its known membrane pore-forming activity (Ragland and Criss, 2017). Lysozyme inhibited the SPI-1 T3SS, which mediates *Salmonella* invasion into host cells when bacteria are extracellular (McGhie et al, 2009). *S.* Typhi in the blood are both extracellular (~37%) and intracellular (~63%) (Wain et al, 1998). Thus, lysozyme has the potential to inhibit invasion but we found no role in cell culture experiments. This was likely due to the 280 ng/ml lysozyme concentrations, which are 100-fold higher in serum and makes the inhibitory mechanism on T3SS possible in vivo. Indeed, toxin-dependent increases in lysozyme relative to that contained in serum significantly inhibited infection. Interestingly, lysozyme is present within phagosomes where it performs its antimicrobial activities against *Staphylococcus aureus* (Shimada et al, 2010). We found lysozyme could localise with intracellular *Salmonella* and inhibit intracellular replication, which is promoted by SPI-1 and is dependent on SPI-2 T3SS virulence effector secretion (McGhie et al, 2009). This mechanism increases the likelihood that lysozyme would interact with *Salmonella* during bacteraemia. Consistent with this, *S.* Typhi encodes a lysozyme inhibitor MliC/YdhA and is likely reprogrammed to express this defence gene during dissemination (Ragland and Criss, 2017). Indeed, deletion of MliC/YdhA significantly inhibits *S.* Typhi invasion and survival in macrophages (Daigle et al, 2001).

Host secretomes undergoing toxin-induced DNA damage were identified by proteomics in the absence of serum (ElGhazaly et al, 2023) that contains abundant plasma proteins, which mask less abundant proteins during proteomics (Geyer et al, 2017). Though we depleted the most abundant plasma proteins in infection challenge samples, our proteomics analysis identified relatively abundant proteins such as APOC3 and lysozyme. This is despite identification of cytokines in the TYGER challenge study investigating typhoid toxin (Gibani et al, 2019), which indicates there are additional unidentified host stress mechanisms to be discovered that suppress the duration of bacteraemia in response to typhoid toxin. Indeed, *Salmonella* expression of lysozyme inhibitors (Ragland and Criss, 2017) makes it unlikely that lysozyme acts alone in counteracting *Salmonella* following DNA damage by the typhoid toxin and opens possibilities for future research. Rather, this study presents the view that pathogen induction of host DNA damage responses elicits antimicrobial responses, which impact infectious disease and explain the shorter duration of bacteraemia in participants infected with wild-type toxigenic *S.* Typhi.

# Methods

## Reagents and tools table

| Reagent/resource | Reference or source | Identifier or catalog number |
|---|---|---|
| **Experimental models** | | |
| Human plasma samples | (Gibani et al, 2019), *deposited at* Oxford Vaccine Centre Biobank | NCT03067961 |
| CACO2 | ATCC | HBT-37 |
| THP1 | ATCC | TIB-202 |
| HCT116 | ATCC | CCL-247 |
| RKO | ATCC | CRL-2577 |
| HepG2 | ATCC | HB-8065 |
| Vi-expressing *S.* Typhi BRD948 *ΔaroA ΔaroC ΔhtrA* | Lowe et al, 1999 | na |
| Vi-deficient *S.* Typhi *ΔtviB ΔaroA ΔaroC ΔhtrA* | Pickard et al, 2008 | na |
| *S.* Javiana | Miller and Wiedmann, 2016 | S5-0395 |
| *S.* Javiana *ΔcdtB* | Miller and Wiedmann, 2016 | M8-0540 |
| *S.* Typhimurium SL1344 | Cain et al, 2004 | na |
| *Escherichia coli* BL21 DE3 | New England Biosciences | C2527 |
| **Recombinant DNA** | | |
| pTrc99A-cdtB | This study | na |
| pM975 | Hapfelmeier et al, 2005 | na |
| pFPV-mCherry | Addgene | 20956 |
| pET-Duet1-pltBHis-pltAMyc-cdtBFLAG-wild-type | Ibler et al, 2019 | plasmid 319 (TxWT) |
| pET-Duet1-pltBHis-pltAMyc-cdtBFLAG-H160Q | Ibler et al, 2019 | plasmid 321 (TxHQ) |
| **Antibodies** | | |
| γH2AX | Millipore | RRID:AB_2755003 |
| APOC3 | GeneTex | RRID:AB_2886149 |
| LYZ | ThermoFisher | RRID:AB_934526 |
| Actin | ThermoFisher | RRID:AB_2536844 |
| GDF15 | Atlas | RRID:AB_1078962 |
| LYZ | ProteinTech | RRID:AB_10639507 |
| DnaK | Novus Biologicals | RRID:AB_11188397 |
| 8-Hydroxydeoxyguanosine | Novus Biologicals | RRID:AB_1260483 |
| Tubulin | Novus Biologicals | RRID:AB_2210209 |
| ISG15 | Santa Cruz | RRID:AB_2126308 |
| p53 | Cell Signalling Technology | RRID:AB_331743 |
| p21 | Cell Signalling Technology | RRID:AB_823586 |
| MAVS | Cell Signalling Technology | RRID:AB_823565 |
| Anti-SopE | Cain et al, 2004 | na |
| Anti-SipB | Hume et al, 2003 | na |
| Anti-FliC | Gerlach et al, 2007 | na |
| Anti-SiiE | Gerlach et al, 2007 | na |
| Alexa-488 donkey anti-mouse | ThermoFisher | RRID:AB_141607 |
| Alexa 594 donkey anti-rabbit | ThermoFisher | RRID:AB_141637 |
| IRDye® 800CW Donkey anti-Mouse | LiCor Biosciences | RRID:AB_621847 |
| RDye® 680RD Donkey anti-Rabbit | LiCor Biosciences | RRID:AB_2716687 |
| **Oligonucleotides and other sequence-based reagents** | | |
| 616_Sty_cdtB_EcoR1_FWD | This study | "Methods" '*Salmonella* infection' |

| Reagent/resource | Reference or source | Identifier or catalog number |
|---|---|---|
| 617_Sty_cdtB_BamH1_Rev | This study | "Methods" 'Salmonella infection' |
| Non-targeting siRNA | Horizon Discovery | D-001810-01-20 |
| LYZ siRNA | Horizon Discovery | L-011079-00-0005 |
| **Chemicals, enzymes and other reagents** | | |
| Multiple Affinity Removal Column Human-14 (MARS-14) | Agilent | 5188-6559 |
| Buffer A | Agilant | 5185-5987 |
| Buffer B | Agilant | 5185-5988 |
| Acetone | Merck | 100014 |
| SDS | Sigma-Aldrich | 05030-500ML-F |
| Tween 20 | VWR | 663684B |
| Non-fat milk powder | Cambridge Biosciences | 54650.1000 |
| MOPS SDS Running Buffer 20x | Fisher Scientific | 13226499 |
| MES SDS Running Buffer 20x | ThermoScientific | NP0002 |
| 40% Acrylamide/Bis Solution, 37.5:1 | Bio-Rad | 1610148 |
| Bis-Tris | Sigma-Aldrich | B9754 |
| Ammonium persulphate | Sigma-Aldrich | A7460 |
| TEMED | ThermoFisher | 17919 |
| UREA | Sigma-Aldrich | U5378 |
| Bromophenol blue | Sigma-Aldrich | 114391 |
| Triethylammonium Bicarbonate Buffer (TEAB) buffer | Thermo Fisher | 90114 |
| PBS | Sigma-Aldrich | P2272 |
| Tris(2-carboxyethyl) phosphine hydrochloride (TCEP) | Merck | C4706-2G |
| Iodoacetamide | Sigma-Aldrich | I6125 |
| HPLC water | Thermo Fisher | W6-1 |
| Phosphoric acid | Fisher scientific | A242- 500 |
| Methanol | Sigma-Aldrich | 900688-1L |
| S-Trap columns | Protifi | C02-micro-80 |
| Trypsin | Thermo Fisher | 90058 |
| Trifluoroacetic acid | Merck | 108262 |
| Acetonitrile | Thermo Fisher | 047138.K2 |
| Formic acid | Thermo scientific | A117-50 |
| DMSO | Sigma-Aldrich | D2438 |
| NiNTA agarose affinity chromatography | Qiagen | 30210 |
| Isopropyl β-D-1-thiogalactopyranoside | Sigma-Aldrich | I5502 |
| Tris-HCl | Sigma-Aldrich | 10812846001 |
| NaCl | Sigma-Aldrich | S7653 |
| MgCl$_2$ | Sigma-Aldrich | M8266 |
| Paraformaldehyde | Thermo Fisher | J61899 |
| Triton-X-100 | VWR Chemicals | 0694-1L |
| Vectashield mounting agent | Vector Lab | H1200 |
| EDTA | Fluorochem | F053299 |
| Glutaldehyde | Merck | G5882 |
| Sodium cacodylate buffer | Clinisciences Limited | 11650 |
| Aqueous osmium tetroxide | Clinisciences Limited | 19160 |
| Aqueous uranyl acetate | BioServ UK Limited | MD16-115 |
| Reynold's lead citrate | Clinisciences Limited | 22410-01 |
| β-Mercaptoethanol | Sigma-Aldrich | M6250 |
| **Cell and bacterial culture** | | |

| Reagent/resource | Reference or source | Identifier or catalog number |
|---|---|---|
| Penicillin/streptomycin | Gibco | 11548876 |
| MEM | Thermo Fisher | 32561037 |
| RPMI-1640 | Sigma-Aldrich | R8758-500ML |
| DMEM | Thermo Fisher | 31966021 |
| McCoy's 5a Medium | ThermoFisher | 16600082 |
| OptiMEM | Gibco | 31985070 |
| Lipofectamine RNAiMax | Thermo Scientific | 13778150 |
| FBS | Sigma-Aldrich | F7524 |
| Trypsin | Sigma-Aldrich | T4049 |
| Phorbol 12-myristate 13-acetate (PMA) | Sigma-Aldrich | P8139 |
| LB Broth | Merck | L3522 |
| Agar | VWR | SIAL05039-500G |
| Sucrose | Merck | 84097 |
| IFN-γ | Merck Millipore | IF002 |
| Lipopolysaccharide | Thermo Fisher | L23352 |
| Hemocytometer | Hawksley | AC1000 |
| 24-well plates | Greiner | G662160 |
| 6-well plates | Greiner | 657160 |
| 0.45 μm filters | Sigma-Aldrich | SLHAR33SS |
| 0.2 μm filters | Sartorius | FIL6720 |
| **Drugs, antibiotics and proteins** | | |
| Cephalexin | Sigma-Aldrich | C4895 |
| Kanamycin | Scientific Laboratory Supplies | 60615 |
| Ampicillin | VWR | A051-B |
| Etoposide | Cayman Chemicals | 12092 |
| Caffeine | Sigma-Aldrich | C0750 |
| CCCP | Selleck Chemicals | S6494 |
| Mitoquinone mesylate | TargetMol | T12059L |
| N-Acetyl-D-cysteine | TargetMol | T38155 |
| Diphenyleneiodonium chloride | TargetMol | T7191 |
| Lysozyme | Merck | L6876 |
| Recombinant lysozyme | Merck | L1667 |
| Lactoferrin | Merck | L9507 |
| IFN-alpha 2 | NKMaxBio | IFN0502 |
| **Kits and fluorescent labels** | | |
| APOC3 ELISA kit | Thermo Fisher | EHAPOC3 |
| LYZ ELISA kit | Abcam | ab108880 |
| Micro BCA Protein Assay | ThermoScientific | 23235 |
| Click-iT EdU Kit for Imaging, Alexa Fluor 647 dye | Thermo Fisher | C10340 |
| Alexa Fluor™ 488 NHS Succinimidyl Ester | ThermoFisher | A20000 |
| Trans-Blot Turbo RTA PVDF Transfer Kit, | Bio-Rad | 1704272 |
| **Software and algorithms** | | |
| Graphpad Prism 9 (9.0.2) | Graphpad by Dotmatics | https://www.graphpad.com |
| Fiji 2.0.0-rc-69/1.52p | ImageJ Wiki | https://ImageJ.net/software/fiji/ |
| Microsoft Excel | Microsoft | https://www.microsoft.com/en-gb/ |
| CellProfiler (4.2.6) | Broad Institute | https://cellprofiler.org |
| Perseus (1.6.10.50) | Max-Planck-Institute of Biochemistry | https://maxquant.net/perseus/ |
| MaxQuant (1.6.10.43) | Max-Planck-Institute of Biochemistry | https://maxquant.net/maxquant/ |
| Adobe Illustrator (29.8.1) | Adobe | www.adobe.com |
| Adobe Photoshop (26.10) | Adobe | www.adobe.com |

| Reagent/resource | Reference or source | Identifier or catalog number |
|---|---|---|
| Image Studio (6.1) | LiCor Biosciences | www.licorbio.com |
| NIS elements software (version 6.10) | Nikon | https://www.microscope.healthcare.nikon.com/en_EU/ |
| **Deposited data** | | |
| Proteomics data | ProteomeXchange Consortium via Pride (https://www.ebi.ac.uk/pride/) | PXD058381 |
| Fiji figure making code | ElGhazaly et al, 2023 | https://doi.org/10.5281/zenodo.8325045 |
| CellProfiler Pipeline | This study | https://doi.org/10.5281/zenodo.17194660 |

## Plasma from human participants

Human blood plasma was obtained from the Oxford Vaccine Centre Biobank with ethical approval from the South-Central Oxford A Ethics Committee in the project entitled 'Investigating Typhoid Fever Pathogenesis (TYGER)' (Ref: 16/SC/0358), clinicaltrials.gov reference NCT03067961 (Gibani et al, 2019). Informed consent was obtained from all human subjects and experiments conformed to the principles set out in the WMA Declaration of Helsinki and the Department of Health and Human Services Belmont Report. Sample size was based on the availability of plasma from bacteraemic participants at the time of typhoid diagnosis following infection with either wild-type (10 samples) or toxin-negative (10 samples) S. Typhi, as well as corresponding baseline samples from uninfected participants (20 samples).

## Immunodepletion of abundant plasma proteins from human participants

Plasma was standardised with respect to protein concentrations using the Micro BCA Protein Assay Kit (Thermo Scientific™, #23235). To immunodeplete the 14 most abundant proteins (albumin, IgG, antitrypsin, IgA, transferrin, haptoglobin, fibrinogen, α2-macroglobulin, α1-acid glycoprotein, IgM, apolipoprotein AI, apolipoprotein AII, complement C3 and transthyretin), plasma (1.5 μg/μl) was diluted 20-fold in Buffer A (proprietary buffer from Agilent; #5188-6559) prior to centrifugation and removal of any particulates (5 min, $16,000 \times g$). Diluted plasma was applied to the Multiple Affinity Removal Column Human-14 (MARS-14, Agilent, #5188-6559). Low-abundance proteins were collected in the flow-through, and high-abundant proteins remained bound to MARS-14 and were eluted in Buffer B (proprietary buffer from Agilent, #5188-6559). Eluted low-abundance proteins were precipitated using acetone to concentrate protein fraction prior to mass spectrometry analysis. Acetone was added to the low-abundance protein elution at $-20\,°C$ (4:1 ratio). Precipitated proteins harvested by centrifugation (10 min, $16,000 \times g$ at 4 °C). Air-dried pellets resuspended in 50 μl of S-Trap solubilization buffer (5% SDS, #05030- 500 ML-F; 5 mM Triethylammonium Bicarbonate Buffer (TEAB); #90114, pH 7.55) and stored at $-20\,°C$ ready for further analysis.

## Sample preparation for mass spectrometry analysis

Samples in S-Trap solubilization buffer were reduced with TCEP (#646547-10X1ML, Sigma-Aldrich) at a final concentration of

10 mM and heated to 70 °C for 15 min. Following this, the samples were incubated with iodoacetamide (#I6125, Sigma-Aldrich) at a final concentration of 10 mM and stored in the dark at room temperature. Phosphoric acid (#A242-500, Sigma-Aldrich) was added to a final concentration of 1.2%. The samples were then diluted in S-Trap binding buffer (90% aqueous methanol, #900688-1 L, Sigma-Aldrich; 0.1 M TEAB, pH 7.1). The diluted samples were loaded onto S-Trap columns (#C02-micro-80, Protifi) by centrifugation (10 s at $4000 \times g$, room temperature) and digested with trypsin. Trypsin (#90058, Thermo Fisher Scientific) in 0.1% trifluoroacetic acid (TFA) (#108262, Sigma-Aldrich) was added at a ratio of 1 μg trypsin per 10 μg of protein immobilized on the S-Trap columns. The S-Traps were sealed with parafilm and incubated at 47 °C for 1 h. Digested proteins were eluted with 50 mM TEAB, followed by centrifugation at $4000 \times g$ for 10 s. Aqueous formic acid was added to the eluate to a final concentration of 0.1% after initially adding 0.2%, and the mixture was centrifuged at $4000 \times g$ for 10 s. Subsequently, 40% of 50% acetonitrile containing 0.2% aqueous formic acid (#A117-50, Sigma-Aldrich) was added, and the samples were centrifuged again under the same conditions. The samples were stored at $-20\,°C$ until they were dried. For drying, they were centrifuged at 45 °C for 90 min in a vacuum at 1400 rpm using an Eppendorf™ Concentrator Plus. Once dried, the samples were resuspended in 0.5% formic acid, transferred to polypropylene vials (Thermo Scientific, #160134), and injected into the Orbitrap for LC-MS/MS analysis.

## LC-MS/MS analysis of proteomic data

Samples were analysed by nanoflow LC-MS/MS using an Orbitrap Elite (Thermo Fisher) hybrid mass spectrometer equipped with an EASY-Spray source, coupled to an Ultimate RSLCnano LC System (Dionex). Xcalibur 3.0.63 (Thermo Fisher) and DCMSLink (Dionex) controlled the system. Peptides were desalted on-line using an Acclaim PepMap 100 C18 nano/capillary BioLC, 100 A nanoViper 20 mm × 75 μm I.D. particle size 3 μm (Fisher Scientific) followed by separation using a 125-min gradient from 5 to 35% buffer B (0.5% formic acid in 80% acetonitrile) on an EASY-Spray column, 50 cm Å ~50 μm ID, PepMap C18, 2 μm particles, 100 °A pore size (Fisher Scientific). The Orbitrap Elite was operated with a cycle of one MS (in the Orbitrap) acquired at a resolution of 60,000 at $m/z$ 400, with the top 20 most abundant multiply charged (2+ and higher) ions in a given chromatographic window subjected to MS/MS fragmentation in the linear ion trap. An FTMS target value of 1e6 and an ion trap MSn target value of 1e4 were used with the lock mass (445.120025) enabled. Maximum FTMS scan accumulation time of 500 ms and maximum ion trap MSn scan accumulation time of 100 ms were used. Dynamic exclusion was enabled with a repeat duration of 45 s with an exclusion list of 500 and an exclusion duration of 30 s.

## MaxQuant analysis of proteomic data

All raw mass spectrometry data were analysed with MaxQuant version 1.6.10.43. Data were searched against a human UniProt sequence database (May 2019) using the following search parameters: digestion set to Trypsin/P with a maximum of two missed cleavages, methionine oxidation and N-terminal protein acetylation as variable modifications, cysteine carbamidomethylation as a fixed modification, match

between runs enabled with a match time window of 0.7 min and a 20-min alignment time window, label-free quantification enabled with a minimum ratio count of 2, minimum number of neighbours of 3 and an average number of neighbours of 6. A first search precursor tolerance of 20ppm and a main search precursor tolerance of 4.5 ppm was used for FTMS scans and a 0.5 Da tolerance for ITMS scans. A protein FDR of 0.01 and a peptide FDR of 0.01 were used for identification level cut-offs.

## Perseus bioinformatic analysis of proteomic data

MaxQuant data output was loaded into Perseus version 1.6.10.50 and all LFQ intensities were set as main columns. The Matrix was filtered removing any contaminants, identified by site and reverse sequences. LFQ intensities were transformed using Log2(x) default function. Rows were then filtered with a minimum value of 5 valid LFQ intensity values in at least one group. Data was evaluated using Pearson correlation analysis and outliers omitted. Sample 8183 D0 was excluded from the analysis due to an inconsistent Pearson correlation value. Missing values were then replaced with a width of 0.3 and down shift 1.8. Sample groups were then compared pairwise with D0 and TD of respective groups using Student $t$ test with permutation-based FDR calculation (FDR = 0.05) with an S0 = 0.1. Data was then exported to Microsoft excel and GraphPad Prism before figure assembly. The mass spectrometry proteomics data have been deposited to the ProteomeXchange Consortium via the PRIDE partner repository with the dataset identifier PXD058381.

## Cell culture

CACO2 (ATCC #HBT-37), HCT116 (ATCC, #CCL-247), RKO (ATCC, #CRL-2577), HepG2 (ATCC #HB-8065) and THP1 (ATCC #TIB-202) cells were stored in cryopreservative media (10% DMSO from Sigma-Aldrich, #D2438) and 90% complete media in liquid nitrogen. Frozen cells were thawed at 37 °C for 90 s and cultured in recommended media supplemented with 1% Penicillin/Streptomycin (Gibco, #11548876), 10% FBS (Sigma-Aldrich, #F7524): CACO2 cells in DMEM from (ThermoFisher, #31966021); HepG2 and RKO cells win MEM (Merck, #M0518), HCT116 in McCoy's 5a Medium (ThermoFisher, #16600082), and THP1 cells in RPMI-1640 (ThermoFischer #R8758-500ML). THP1 differentiation from monocytes into M1 macrophages was performed by addition of 10 ng/ml Phorbol 12-myristate 13-acetate (PMA, Merck, #P1585) for 48 h. PMA was replaced with in complete media containing 50 ng/ml IFN-γ (Merck Millipore, #IF002) and 15 ng/ml lipopolysaccharide (Thermofisher, #L23352) for 24 h. Prior to experiments, IFN-γ and LPS were removed. All cell lines were cultured at 37 °C in a humidified incubator with 5% $CO_2$.

Cells were passaged every 3 to 5 days depending on their doubling time. THP1 monocyte cells grown in suspension and were split by diluting cell cultures. For sub-culturing adherent cells, trypsin (Sigma-Aldrich, #T4049) was used to detach cells followed by neutralisation with FBS containing media. Cell viability was determined using trypan-blue (Sigma-Aldrich, #T8154) followed by quantification of viable (trypan-blue negative) and non-viable (trypan blue-positive) cells using a hemocytometer (Hawksley, #AC1000). For immunofluorescence studies, cells were seeded in

24-well plates (Greiner #G662160) containing glass coverslips (VWR, #631-1578) using complete growth media, or in the absence of coverslips for quantification of *Salmonella* CFUs following infection. For immunoblotting experiments, cells were seeded into 6-well plates (Greiner, #657160). If necessary, conditioned media was harvested by centrifugation at $6000 \times g$ for 5 min to pellet the cells and cell debris removed by filtering through 0.2 μm filters (Sartorius, #FIL6720) and storage at −80 °C.

## Recombinant toxin purification and intoxication assays

The typhoid toxin was purified from *Escherichia coli* BL21 DE3 (NEB, #C2527) pETDuet-1 encoding $pltB^{His}$ $pltA^{Myc}$ and $cdtB^{FLAG}$ using NiNTA agarose (Qiagen) affinity chromatography according to manufacturer instructions as previously described (Ibler et al, 2019). Unless stated otherwise, cells were intoxicated with 20 ng/ml toxin (~175 picomolar) for 2 h, washed three times with sterile PBS (Sigma-Aldrich, #J60801.K3) to remove any extracellular toxin and chase with fresh complete growth media for the duration of the experiment.

### *Salmonella*

Serovars of *Salmonella enterica* in the study were maintained on LB agar plates and cultured in LB broth. *S*. Javiana (S5-0395) and Δ*cdtB* (M8-0540) (Miller and Wiedmann, 2016) were kind gifts from Prof. Martin Weidmann (New York). Vaccine candidate *S*. Typhi Ty2 BRD948 Δ*aroA* Δ*aroC* Δ*htrA* (Lowe et al, 1999), and a Vi-deficient *tviB* null mutant derivative strain (Pickard et al, 2008) were kind gifts from Prof. Gordon Dougan (Cambridge). *S*. Typhimurium SL1344 was a kind gift from Prof. Vassilis Koronakis (Cambridge).

### *Salmonella* infection

*S*. Javiana wild-type or Δ*cdtB* encoding pM975, which expresses GFP when bacteria are intracellular (Hapfelmeier et al, 2005) were cultured in LB 50 μg/ml ampicillin at 37 °C in a shaking incubator to 2.0 OD$_{600}$. To complement the Δ*cdtB* mutation, *S*. Javiana was transformed with pTrc99A encoding cdtB (cloned EcoR1/BamH1 using the primers 616_Sty_cdtB_EcoR1_FWD CCCCGAATT-CATGTTAAGACACATTCAAAATAG; 617_Sty_cdtB_BamH1_-Rev GGGGGGATCCTTAACAGCTTCGTGCCAAAAAGGCTAC). The multiplicity of infection (MOI) was optimised for CACO2 and HCT116 cells (MOI 100). To assay *Salmonella*-induced host cell signalling, infection was initiated in the absence of antibiotics by addition of *Salmonella* to cell cultures in complete growth medium and centrifugation for 1 min at $1000 \times g$ followed by 30 min incubation at 37 °C 5% $CO_2$. Infected cells were washed three times with PBS and incubated in growth media containing 50 μg/ml gentamicin (Chem Cruz, sc203334) for 1.5 h then reduced to 10 μg/ml gentamicin for the rest of the experiment. When assaying *Salmonella* invasion, the method was modified by serum-starving cells 24 h prior to infection that deprives cells of membrane ruffling stimulants in FBS. To assess infection efficiency, *Salmonella* invasion occurred over 30 min. After 1.5 h incubation with 50 μg/ml gentamicin (to assay invasion) or 24 h incubation with 10 μg/ml gentamicin (to assay intracellular infection), cells were washed three times with PBS and lysed with 1% Triton X-100.

Serial dilutions of cell lysates were used to inoculate (5 μl) LB agar plates containing 50 μg/ml ampicillin and the *Salmonella* cultured overnight at 37 °C. *Salmonella* colony counts were used to quantify colony forming units (CFUs).

## siRNA transfection

Per well of a 24-well plate format, 0.5 μl Lipofectamine RNAiMax (Thermo Scientific #13778150) and non-targeting siRNA (Horizon Discovery, #D-001810-01-20) or LYZ siRNA (Horizon Discovery, #L-011079-00-0005) were prepared (0.5 μl of 20 μM stock) in two different tubes of 25 μl OptiMEM media (Gibco, #31985070), then mixed together for 5 min at room temperature. The 50 μl mix of siRNA and lipofectamine were added to 450 μl complete growth media and incubated on HCT116 cells for 48 h. The final concentration of siRNA in culture was 20 nM. At 48 h, cells were intoxicated for 2 h followed immediately by 72 h incubation 37 °C, 5% $CO_2$. At 72 h, cells were either used for immunoblotting or infected with *S.* Javiana and infection efficiency quantified by calculating CFUs on LB agar plates.

## Treatment with APOC3, lysozyme or interferon

Purified APOC3 (Novus Biologics; NBP1-99294) at 50 mg/ml were diluted with bacteria in M9 minimal media. Endogenous conventional LYZ (Sigma-Aldrich, #L2879-1G) or recombinant human conventional LYZ (Sigma-Aldrich, #L1667) were resuspended in 10 mM Tris-HCl pH 8 at 10 mg/ml. *Salmonella* was grown in either LB or M9 minimal media before addition of LYZ at concentrations spanning 0.1 μg/ml to 1000 μg/ml as indicated accordingly in figure legends. For denatured LYZ, 10 mg/ml LYZ in 10 mM Tris-HCl pH 8 was incubated at 95 °C for 15 min and used at 100 μg/ml when treating *Salmonella*. For generating fluorescent LYZ, 10 mg/ml LYZ in PBS was spiked with Alexa Fluor™ 488 NHS Succinimidyl Ester (ThermoFisher, #A20000) on the end of a p20 tip. The mixture was incubated in the dark at room temperature for 10 min before quenching the fluorophore by diluting the mixture 10-fold in the amine-containing buffer TBS pH 7.4 to generate a 1000x stock for experiments. Recombinant human IFN-alpha 2 (NKMaxBio, #IFN0502) was used to treat HCT116 cells by adding to cultured media at 100 ng/ml for the duration of the experiment.

## Immunoblotting

For immunoblotting *Salmonella* whole cell lysates and secreted proteins, bacteria were cultured from 0.2 $OD_{600}$ in LB broth in the presence or absence of LYZ (0.1–1000 μg/ml), recombinant LYZ (1000 μg/ml), LFN (100 μg/ml) or EDTA (1 mM) for 2 h. To generate whole cell lysates, *Salmonella* were harvested at 8000 RCF for 10 min and the bacterial pellet resuspended in SDS-UREA (50 mM Tris-HCl pH 6.8, 2% SDS, 6 M UREA, 0.3% Bromophenol Blue) containing 5% β-Mercaptoethanol (Sigma-Aldrich, #M6250) according to their $OD_{600}$ (100 μl SDS-UREA per OD unit). To harvest supernatants, culture supernatant was re-centrifuged and the supernatant filter sterilised using 0.45 μm filters (Sigma-Aldrich, #SLHAR33SS) before adding 10% v/v trichloroacetic acid (TCA from Sigma-Aldrich, #91228) to precipitate proteins overnight at 4 °C. Precipitated proteins were harvested by centrifugation at 10,000 RCF, 30 min at 4 °C. Supernatant was discarded and precipitated proteins washed in 100% acetone by centrifugation at

10,000 RCF, 30 min at 4 °C. Air-dried precipitated proteins were resuspended in SDS-UREA according to their $OD_{600}$ and tenfold more concentrated than whole cell lysates (10 μl SDS-UREA per OD unit). To analyse the export of flagellin, *S.* Typhimurium SL1344 were cultured to 0.5 $OD_{600}$ in LB broth before applying shear force (passed through a 25 gauge needle 5 times then vortexed at 1000 rpm for 20 min). Whole lysates were prepared −/+ shear force (start point at 0 h), and *S.* Typhimurium were cultured with or without LYZ for 2 h before preparing whole cell lysates. For immunoblotting mammalian cells, cells were scraped into 1.5 ml tubes using 1 ml PBS and cell scrapers (VWR; #734-2602) at indicated timepoints, and whole cell lysates generated by re-suspending cultured cells in SDS-UREA according to their $OD_{600}$ (200 μl SDS-UREA per OD unit).

Proteins were separated by 9% or 12% Bis-Tris SDS-PAGE gels using MOPS (high molecular weight protein separation) or MES (low molecular weight) buffer (50 mM MOPS or MES, 50 mM Tris, 0.1% SDS, 20 mM EDTA) and transferred to PVDF membrane (#1704272, Bio-Rad) using a Trans-Blot Turbo Transfer System (Bio-Rad). PVDF membranes were blocked for 1 h with TBS-Tween 5% milk (Tris pH 7.4, 100 mM NaCl, 0.1% Tween® 20, 5% non-fat dried milk). Primary antibody incubations (1 h to 24 h) and washes were performed in TBS-Tween. IRDye-labelled secondary antibodies were incubated in TBS-Tween 5% milk (30 min). Immunoblot images were captured using Odyssey Sa (LiCor) and the software Image Studio 6.1 (LiCor) before exporting TIFF files for figure assembly.

## Antibodies for immunoblotting and immunofluorescence

Antibodies were purchased from Millipore (γH2AX #05-636-I, RRID:AB_2755003, diluted 1:1000), GeneTex (APOC3 #GTX129994, RRID:AB_2886149, 1:250), ThermoFisher, (LYZ #MA1-82873, RRID:AB_934526, 1:250; actin #MA1-140, RRID:AB_2536844, 1:1000), Atlas (GDF15 #HPA011191, RRID:AB_1078962, 1:500), ProteinTech (LYZ #15013-1-AP, RRID:AB_10639507, 1:250), Novus Biologicals (DnaK #NBP1-97490, RRID:AB_11188397, 1:500; 8-Hydroxydeoxyguanosine #NB110-96878, RRID:AB_1260483, 1:1000; tubulin #NB100-690, RRID:AB_2210209, 1:500), Santa Cruz (ISG15 sc-166755, RRID:AB_2126308, 1:1000) and Cell Signalling Technology (p53 #2524, RRID:AB_331743, 1:250; p21 #2947, RRID:AB_823586, 1:1000; MAVS #3993, RRID:AB_823565 1:1000). *Salmonella* antibodies raised in rabbit were anti-SipB (Hume et al, 2003) and anti-SopE (Cain et al, 2004), which were kind gifts from Prof. Vassilis Koronakis (University of Cambridge, UK). Also, anti-FliC and anti-SiiE (Gerlach et al, 2007), which were kind gifts from Prof. Michael Hensel (Universität Osnabrück, Germany). For immunofluorescence, we used secondary antibodies (diluted (1:1000) from ThermoFisher Scientific (Alexa-488 donkey anti-mouse IgG, #A-21202, RRID:AB_141607; Alexa 594 donkey anti-rabbit IgG, #A-21207, RRID:AB_141637). For immunoblotting, we used secondary antibodies (1:10,000) from LiCor Biosciences (IRDye® 800CW Donkey anti-Mouse IgG, #926-32212, RRID:AB_621847; IRDye® 680RD Donkey anti-Rabbit IgG, #925-68073, RRID:AB_2716687).

## Cephalexin treatment

*S.* Javiana pFPV-mCherry (Drecktrah et al, 2008), *S.* Typhi BRD948 or *S.* Typhimurium SL1344 were cultured in LB broth until an $OD_{600}$ of 0.7. Bacteria were diluted 1:10 and grown in LB broth

with 50 μg/ml Cephalexin for 2.5 h at 42 °C, 120 rpm. Culture was diluted 1:10 in LB Broth supplemented with 0.8 mM IPTG for 1 h at 37 °C, 120 rpm. Bacteria were harvested by centrifugation and resuspended in 1 M sucrose on ice. 1 ml *Salmonella* suspensions were incubated with 80 μl of Tris-HCL pH 8 with or without addition of LYZ (1 mg/ml), EDTA (1 mM) or LFN (100 μg/ml) at room temperature for 20 min. Samples were fixed using PBS 4% paraformaldehyde and air-dried on glass coverslips ready for imaging by microscopy.

## Immunofluorescence microscopy

To assay cell cycle arrest, EdU was added to the culture 24 h before fixation and we used Click-iT™ EdU Cell Proliferation Alexa FluorTM 647 Kit for imaging according to manufacturer instructions (Thermofisher, #C10340). At experimental endpoints, CACO2, HCT116 or HepG2 cells were washed with PBS (Biotech, PD8117) then fixed with PBS 4% paraformaldehyde (PFA) (ThermoFisher, #J61899) for 15 min at room temperature. Cells were washed two more times with PBS then blocked and permeabilised using blocking buffer, namely PBS 3% BSA (Sigma-Aldrich, #1073508600) and 0.2% Triton X-100 (VWR, #28817.295) at room temperature for 1 h. Primary and secondary antibodies were incubated with cells in blocking buffer consecutively for 1 h and 30 min, respectively, then washed with PBS then water and left to air-dry. Coverslips were mounted and counterstained on 6 μl of VectaShield mounting agent with DAPI (Vector Lab, #H1200), and sealed with nail varnish before being imaged on Nikon's Inverted Ti Eclipse equipped with an Andor Zyla sCMOS camera (2560 × 2160; 6.5 μm pixels). The objectives used were Plan Apo 10x (NA 0.45); Plan Apo 20x (NA 0.75); Plan Fluor 40x oil (NA 1.3); Apo 60x oil (NA 1.4); Plan Apo 100x Ph oil (Na 1.45); Plan Apo VC 100x oil (NA 1.4). Quad emission filters for used with SpectraX LED excitation (395 nm, 470 nm, 561 nm, 640 nm). The imaging software used was NIS elements software (version 6.10).

## Electron microscopy

Specimens were fixed overnight in a solution of 2.5% glutaldehyde in 0.1 M sodium cacodylate buffer. The following day excess fixative was removed and washed free of excess fixative in two changes of 10 min each in 0.1 M sodium cacodylate buffer. Samples were then post-fixed 2% aqueous osmium tetroxide for 2 h. Samples were washed free of excess Osmium tetroxide in buffer solution and dehydrated through a graded series of ethanol solutions in water (50%, 75%, 95%, 100% twice and dried 100% ethanol), and cleared of excess ethanol in epoxypropane (EPP) and then infiltrated in a 50/50 araldite resin: EPP mixture overnight on a rotor. The next day this mixture was replaced with two changes over 8 h of fresh araldite resin mixture before being embedded into EM moulds in fresh araldite resin and cured in a 60 °C oven for 48-72 h. Ultrathin sections, approximately 85 nm thick, were cut using a Diatome histoknife on a Reichart Jung ultracut E ultramicrotome and picked up onto formvar-coated 200 mesh copper grids. These were stained for 30 min with saturated aqueous uranyl acetate, blotted free of excess stain using filter paper and washed in distilled water. Sections were counterstained in Reynold's lead citrate, washed in water again and blotted dry. Sections were examined using an FEI

Tecnai Spirit Biotwin 120 Transmission Electron Microscope at an accelerating voltage of 80Kv. Electron micrographs were recorded using Gatan Orius 1000 digital camera and Gatan Digital Micrograph software.

## ELISA

APOC3 (ThermoFisher, #EHAPOC3) and LYZ ELISA kits (Abcam, #ab108880) were used as per manufacturer instructions. Human Plasma samples were diluted 1:500 in an assay buffer and incubated on pre-coated ELISA plates overnight at 4 °C. Conditioned media was collected from cultured cells, centrifuged at 6000xg for 5 min to remove any dead cells or cell debris, and the supernatant was passed through 0.2 μm filters. The filtrate was subsequently serially diluted fivefold to optimise assay sensitivity and incubated on pre-coated ELISA plates overnight at 4 °C. The supernatant was removed and bound proteins washed using proprietary buffer before addition of biotin-labelled primary antibody conjugate to APOC3 or LYZ for 1 h at room temperature. Streptavidin-HRP (horseradish peroxidase) was incubated for 1 h at room temperature followed by tetramethylbenzidine substrate of HRP for 30 min. Plates were imaged on FLUOstar Omega at 450 nm absorbance and a standard curve was generated, sample concentration was extrapolated from standard curve and normalised to cell density.

## Drugs

Drugs were resuspended as per manufacturer's instructions and diluted in media to their working concentrations. Drugs were obtained from the following sources and used at indicated concentrations: 10 μM etoposide (Cambridge Bioscience, #1209), 10 mM caffeine (Tocris, #2793/100), 10 μM carbonyl cyanide 3-chlorophenylhydrazone (Selleck Chemicals, #S6494), 10 μM mitoquinone mesylate (Cambridge Bioscience, #T12059L), 1 μM N-Acetyl-D-cysteine (TargetMol, #T38155) and 10 μM diphenyle-neiodonium chloride (TargetMol, #T7191). Drugs were added to cultured cells after 2 h treatment with typhoid toxin variants was performed and incubated for the duration of the experiment.

## Image processing

Fiji 2.0.0-rc-69/1.52p macro code was created to automate image processing and generation of figure panels (https://doi.org/10.5281/zenodo.8325045), as previously described (ElGhazaly et al, 2023). Briefly, brightness and contrast are normalised across all images of interest, a pre-set ROI is used to crop images to regions of interest, and a pre-set scale bar is added. DAPI nuclei are then outlined, overlaid with the other channels and saved as png files. Many variants of the code have been created to cater for 2-channel, 3-channel and 4-channel images, with either normal composite images or with DAPI outlines. Pseudo colours were assigned to fluorescence micrographs to make interpretation more accessible for those with colour blindness.

## Quantification of fluorescence microscopy images

Microscopy image analysis was carried out using CellProfiler (version 4.2.6) (Carpenter et al, 2006) using the following

CellProfiler pipeline: https://doi.org/10.5281/zenodo.17194660. Nuclei were segmented using *IdentifyPrimaryObjects* (DAPI). Nuclear intensity of γH2AX and EdU was quantified using *MeasureObjectIntensity* and later processed in Excel. Whole-image intensity of non-nuclear markers (APOC3 and LYZ) was measured using *MeasureImageIntensity*. Cell boundaries were approximated using *IdentifySecondaryObjects*, expanding ~50 pixels from the nuclear boundary defined in *IdentifyPrimaryObjects*. The *Threshold* module was used to generate binary masks, which were applied with *MaskObjects* to restrict quantification to the secondary objects. Cells were then classified as positive or negative for marker expression using *ClassifyObjects*. Quantitative data were exported via *ExportToSpreadsheet* and further analysed in Excel. Threshold intensity cut-offs for γH2AX and EdU were determined in Excel; nuclei with intensity values above these thresholds were scored as positive. The number of positive nuclei were summed, and percentages calculated.

## Statistical analysis

Statistical analysis was carried out by Graphpad Prism 9 version 9.0.2. Statistical analysis was performed using Welch's *t* test, or ANOVA followed by appropriate post-tests to control for multiple testing between groups, as indicated in figure legends. The significance is represented using *$P < 0.05$ (*), $P < 0.01$(**), $P < 0.001$ (***), and $P < 0.0001$(****). The raw data used for presenting quantitative data is provided (first tab of excel files). Source data files also provide additional statistical information indicating the normal distribution of data calculated by Bartlett"s and Brown–Forsythe's tests, and mean ± Standard Deviation ( ± SD) that provides an estimate of within-group variation (second tab).

## Study design

To minimise subjective bias, fluorescence microscopy images were analysed using CellProfiler or Fiji with predefined, rule-based image-processing algorithms to ensure objective and reproducible quantification of phenotypes. Images were acquired at five predefined fields of view per coverslip across all experiments to standardise sampling. Randomisation of treatments and blinding of image acquisition/analysis were not performed. Inclusion criteria were pre-established, requiring that all experimental data be validated by both positive and negative internal controls. Datasets were excluded only in cases where control validation was not possible due to attrition (e.g., technical failure). No intentional exclusions of data points were made. All criteria for inclusion and exclusion were defined prior to data analysis.

# Data availability

The proteomics data in this study are available on the ProteomeXchange Consortium via the PRIDE partner repository (http://www.ebi.ac.uk/pride) with the dataset identifier PXD058381.

The source data of this paper are collected in the following database record: biostudies:S-SCDT-10_1038-S44321-025-00347-8.

## The paper explained

### Problem
It is well established that the DNA damage response (DDR) protects humans against cancer but its role in defence against bacterial pathogens is less clear. Acute typhoid fever is caused by *Salmonella* Typhi, which induces DNA damage through typhoid toxin that was examined in human participants infected with *Salmonella*. Surprisingly, the presence of typhoid toxin reduced the duration of symptoms such as blood infection, which suggests DDRs protect humans against typhoid fever. The mechanisms are unknown.

### Results
We mapped the plasma proteome in the blood of human participants infected with wild-type or toxin-negative *S.* Typhi. This showed that typhoid toxin triggers release of the antimicrobial enzyme lysozyme that degrades bacterial cell walls. Lysozyme expression was mediated by activation of the tumour suppressor p53. This followed toxin-induced oxidative stress in mitochondria that damaged DNA in the nucleus. Lysozyme disabled the ability of *Salmonella* to secrete virulence effector proteins and inhibited intracellular infections.

### Impact
The DDR protects humans from cancer. The results presented here provide evidence that the DDR has co-evolved to defend humans against important bacterial infections such as typhoid fever.

# Peer review information

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

## Acknowledgements

This work was supported by the Human Infection Challenge Network for Vaccine Development (HIC-Vac) funded by the GCRF Networks in Vaccines Research and Development which was co-funded by the MRC and BBSRC (MR/R005982/1). This UK funded award is part of the EDCTP2 programme supported by the European Union. The research was funded by a UKRI Future Leaders Fellowship to DH (MR/S034390/1, MR/X02329X/1) and supported by Medical Research Council [MR/N013840/1] PhD studentship to FB and JWS. We would like to thank the services provided by core facilities at the University of Sheffield: proteomics was performed at the biOMICS Facility (UKRI funding MR/X012220/1, BB/Z515796/1) and fluorescence microscopy performed at the Wolfson Light Microscopy Facility (UKRI funding MR/K015753/1). We would like to thank Prof. Vassilis Koronakis (University of Cambridge) for key antibodies and *S.* Typhimurium, Prof. Martin Wiedmann (Cornell) for *S. Javiana* strains, Prof. Gordon Dougan (Cambridge) for *S.* Typhi strains and Dr. Chris Hill for assistance with electron microscopy (University of Sheffield).

## Author contributions

**Salma Srour**: Data curation; Formal analysis; Validation; Investigation; Methodology; Writing—original draft; Writing—review and editing. **Francesca K Brown**: Formal analysis; Investigation. **James W Sheffield**: Formal analysis; Investigation. **Mohamed ElGhazaly**: Formal analysis; Investigation; Writing—original draft; Writing—review and editing. **Daniel O'Connor**: Resources; Funding acquisition; Writing—original draft. **Malick M Gibani**: Resources; Funding acquisition; Writing—original draft. **Thomas C Darton**: Funding acquisition; Writing—original draft. **Andrew J Pollard**: Resources; Funding acquisition; Writing—original draft. **Mark O Collins**: Data curation; Formal analysis; Supervision; Funding acquisition; Validation; Investigation; Methodology; Writing—original draft; Writing—review and editing. **Daniel Humphreys**: Conceptualization; Formal analysis; Supervision; Funding acquisition; Validation; Investigation; Methodology; Writing—original draft; Project administration; Writing—review and editing.

Source data underlying figure panels in this paper may have individual authorship assigned. Where available, figure panel/source data authorship is listed in the following database record: biostudies:S-SCDT-10_1038-S44321-025-00347-8.

## Disclosure and competing interests statement

The authors declare no competing interests.

# Expanded View Figures

**Figure EV1.  APOC3 expression in HepG2 liver cells treated with typhoid toxin.**

(A) Fluorescence microscopy images of HepG2 intestinal cells, from three independent experiments, either untreated, treated with wild-type typhoid toxin (TxWT) or H160Q DNase-deficient toxin (TxHQ) for 2 h prior to imaging at 96 h of EdU (magenta) or APOC3 (yellow). DAPI-stained nuclear outlines shown. Scale bars: 50 μm. (B) Bar chart showing proportion of APOC3-positive cells ($n = 4$), or (C) EdU-positive HepG2 cells ($n = 3$), at 96 h. Circles indicate biological repeats. (D) Quantification of *Salmonella* CFUs following incubation with 50 mg/ml of purified APOC3 at 1 h, 2 h and 4 h ($n = 3$). Statistical significance: Welch's unpaired t-test for paired measures with unequal variances in (B, C), two-way ANOVA Sidak multiple comparisons (D) assessing 2 independent variables. Data are presented as mean ± SEM. Asterisks indicate significance: *$P < 0.05$, **$P < 0.01$, ***$P < 0.001$, ****$P < 0.0001$. No significance (ns). Exact $P$ values in Appendix Table S1. Circles and n represent biological replicates. Experiments in EV2 linked to Fig. 2. Source data are available online for this figure.

A

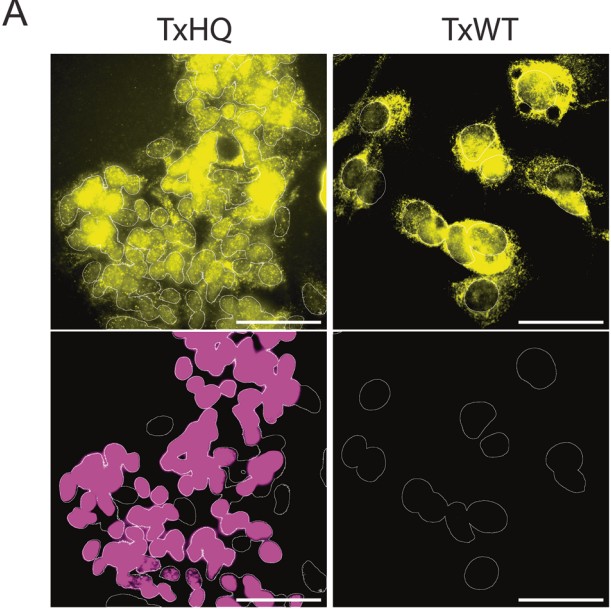

B

C

D

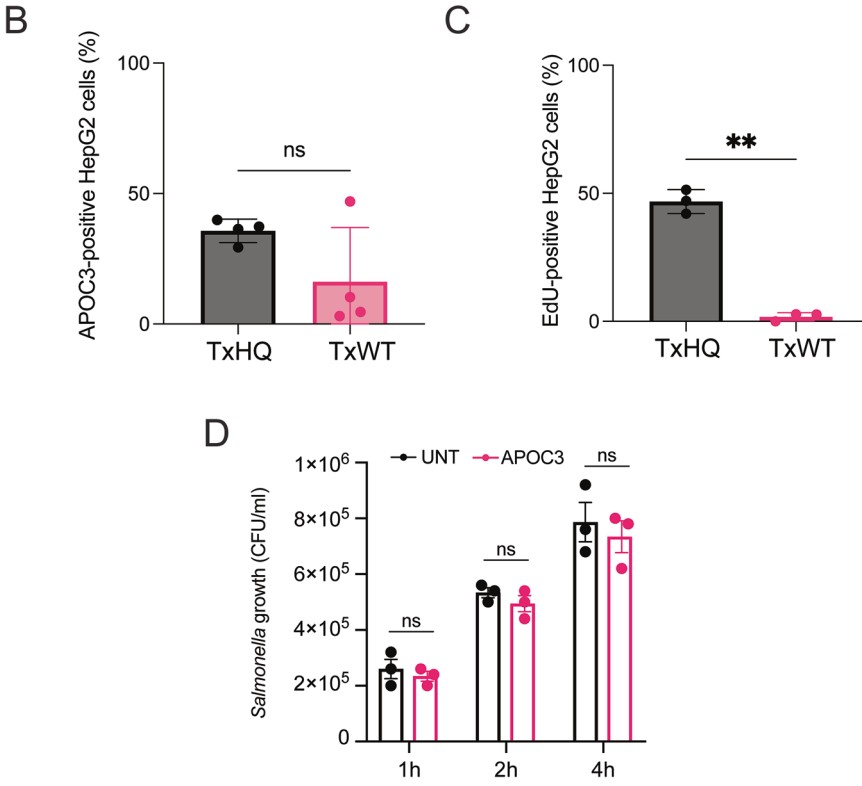

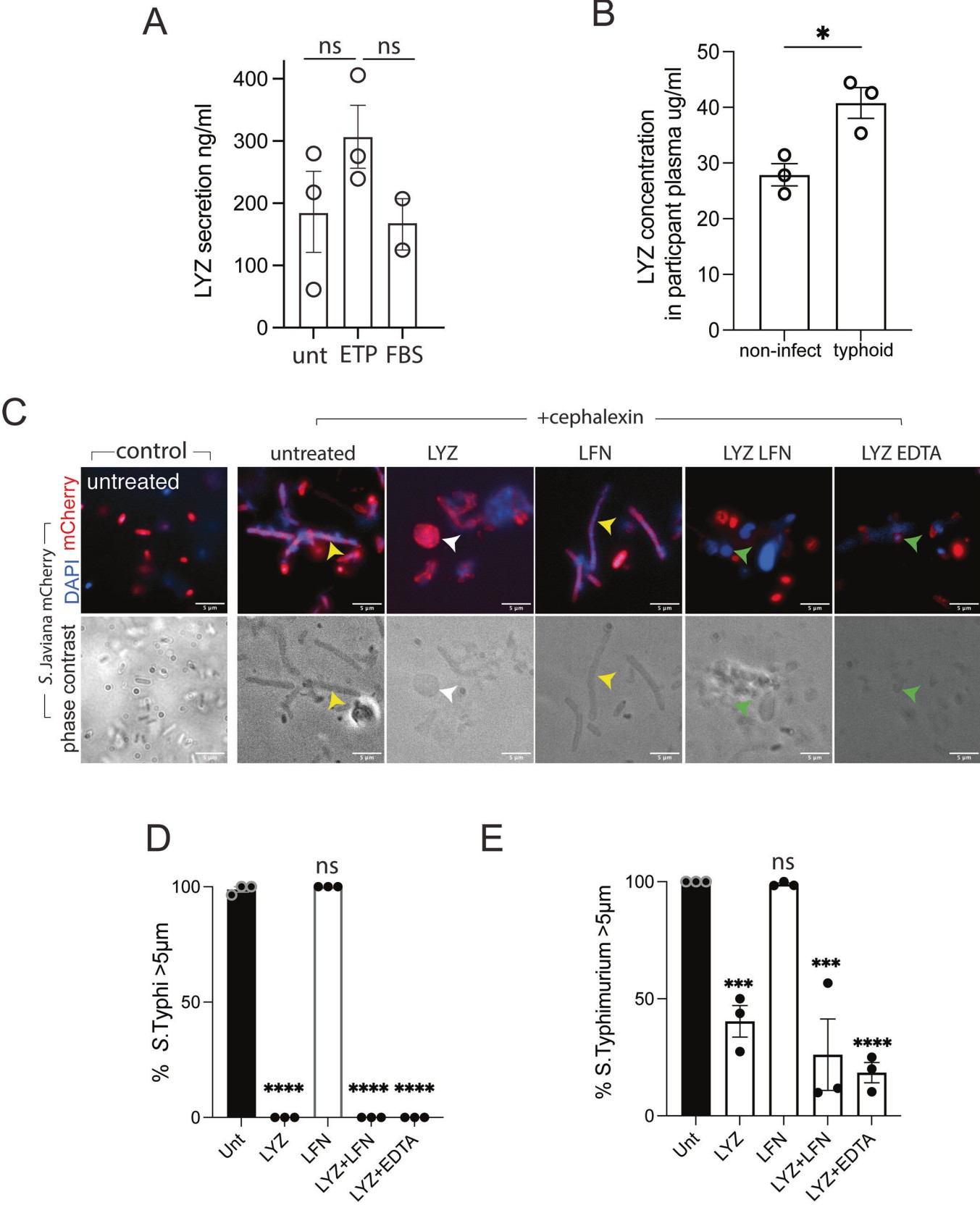

◀ **Figure EV2. Spheroplast formation in response to LYZ.**

(A) ELISA of LYZ using growth media from untreated or etoposide-treated (ETP) CACO2 cells at 96 h, or 10% FBS used to supplement growth media as control ($n = 3$). (B) ELISA of LYZ using plasma from human participants in TYGER study at baseline (non-infect) or at typhoid diagnosis following *S.* Typhi infection (typhoid). Circles represent participants and biological repeats ($n = 3$). (C) Fluorescence microscopy images of cephalexin-treated *S.* Javiana pFPV-mCherry treated with LYZ, LFN, LYZ-LFN or LYZ-EDTA, from three independent experiments, before imaging mCherry *Salmonella* and DAPI-staining by fluorescence microscopy (top panel) or phase contrast (bottom panel). Elongated bacteria >5 μm (yellow arrows), spheroplasts with loss of mCherry (green arrows), and spheroplasts with mCherry retention (white arrows). Untreated, LYZ, and LYZ/LFN images reused in Fig. 3G. Scale bars: 5 μm. Bar charts showing the proportion of long bacteria (>5 μm) following cephalexin treatment of (D) *S.* Typhi or (E) *S.* Typhimurium in the absence (unt) or presence of LYZ, LFN, LYZ and LFN, LYZ and EDTA ($n = 3$). Statistical significance: one-way ANOVA Tukey's multiple comparison (A) analysing all pairs of >3 groups; Welch's unpaired *t* test for paired measures with unequal variances in (B); one-way ANOVA with Brown–Forsythe (D, E) for unequal variances (>3 groups). Data are presented as mean ± SEM. Asterisks indicate significance: *$P < 0.05$, **$P < 0.01$, ***$P < 0.001$, ****$P < 0.0001$. No significance (ns). Exact *P* values in Appendix Table S1. Circles represent biological replicates. Experiments in EV2 linked to Fig. 3. Source data are available online for this figure

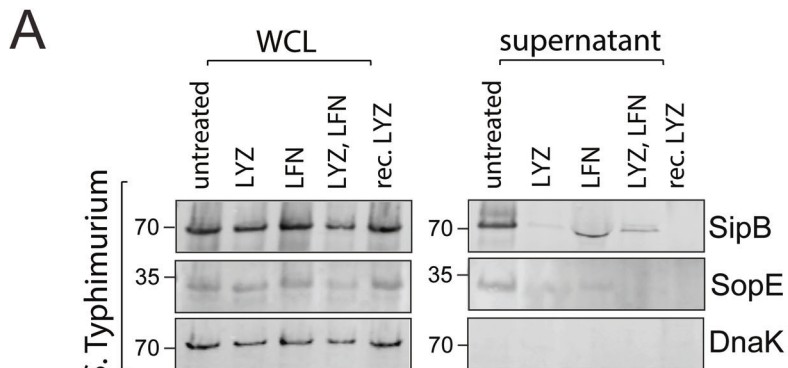

◀ **Figure EV3.** Influence of LYZ on the T3SS of *Salmonella enterica.*

(A) Immunoblot of *S.* Typhimurium in LB broth either untreated, or cultured with 1 mg/ml endogenous LYZ, 100 µg/ml LFN, LYZ and LFN, or 1 mg/ml recombinant LYZ (rec. LYZ) for 2 h ($n = 2$). Whole cell lysates (WCLs) or supernatants immunoblotted with antibodies to virulence effectors SipB or SopE, or the intracellular loading control DnaK. MW in kDa, left. Whole cell lysates or supernatants from (B) *S.* Typhi BRD948 or (C) mutant Vi-deficient *S.* Typhi BRD948, cultured in LB only (untreated) or treated for 2 h with indicated concentrations of LYZ ($n = 2$). Antibodies to SipB or DnaK indicated. MW in kDa, left. (D) Export of *S.* Typhimurium FliC (Flagellin) to the outer membrane in the presence of indicated LYZ concentrations. *S.* Typhimurium were cultured to 0.5 $OD_{600}$ in LB then either left untreated (−) or subjected to shear forces (+) to break flagella before incubating for 2 h with indicated concentrations of LYZ ($n = 2$). Whole cell lysates were immunoblotted with antibodies to flagellin or DnaK. MW in kDa, left. (E) *S.* Javiana (SJ) CFUs calculated on LB agar plates at 24 h post-infection from HCT116 cells already treated for 72 h with TxWT or TxHQ, ($n = 3$). (F) Immunoblot showing LYZ knockdown. HCT116 cells were transfected with non-targeting (NT) or LYZ siRNA for 48 h prior to treatment with TxWT and immunoblotting after 48 h (96 h total) ($n = 2$). Whole cell lysates immunoblotted with LYZ or actin antibodies. MW in kDa, left. Statistical significance: Welch's unpaired *t* test for paired measures with unequal variances in (E). Data are presented as mean ± SEM. Asterisks indicate significance: *$P < 0.05$, **$P < 0.01$, ***$P < 0.001$, ****$P < 0.0001$. No significance (ns). Exact *P* values in Appendix Table S1. Circles and n represent biological replicates. Experiments in EV2 linked to Fig. 4. Source data are available online for this figure

A

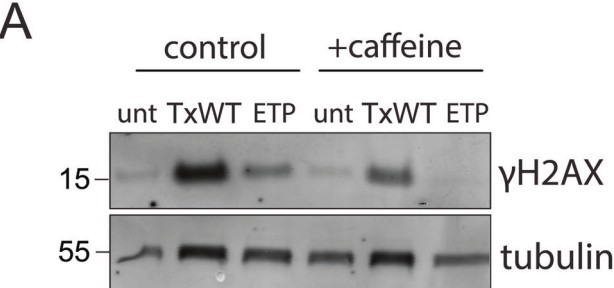

B

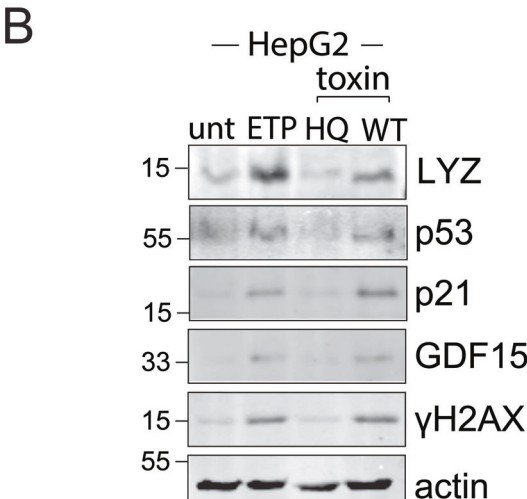

C

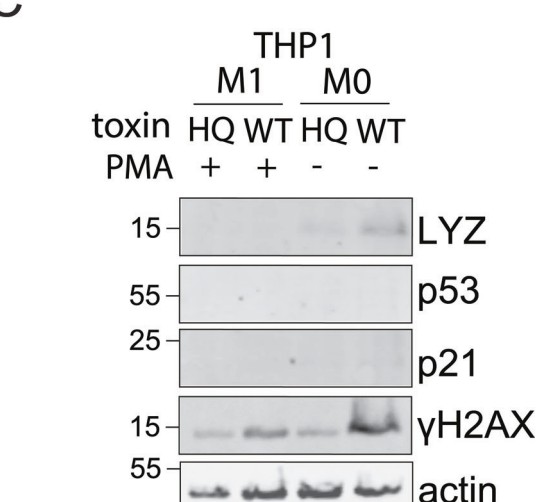

D

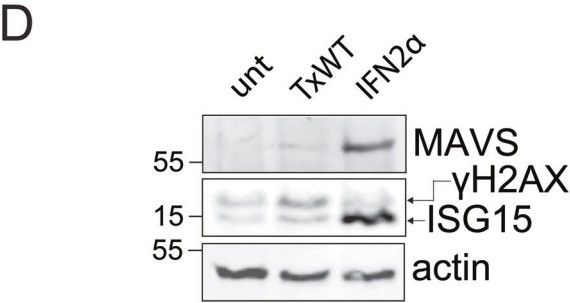

**Figure EV4.   LYZ expression in response to DDRs.**

(A) Immunoblot showing the effect of caffeine on DDRs induced by TxWT or ETP. CACO2 cells were treated for 2 h with TxWT or ETP before addition of 10 mM caffeine for 48 h ($n = 2$). Whole cell lysates immunoblotted with antibodies to γH2AX and tubulin. MW in kDa, left. (B) LYZ expression and p53 responses in HepG2 liver epithelial cells at 72 h following no treatment (unt), or treatment with etoposide, TxHQ or TxWT ($n = 2$). Immunoblots performed with indicated antibodies. MW in kDa, left. (C) The same experiment as (B) performed with TxHQ or TxWT in THP1 macrophages differentiated to non replicate with addition of PMA (+), or in the precursor replicating THP1 monocyte form ($n = 2$). (D) Immunoblotting of interferon-stimulated genes MAVS and ISG15 in HCT116 cells at 72 h when untreated (unt), treated with TxWT or IFN2α as control ($n = 2$). Antibodies indicated, right. MW in kDa, left. Experiments in EV4 linked to Figs. 5 and 6. Source data are available online for this figure

