## [Peer Review File · EMBO Molecular Medicine]

Typhoid toxin of *Salmonella Typhi* elicits host antimicrobial response during acute typhoid fever

Salma Srour, Francesca Brown, James Sheffield, Mohamed ElGhazaly, Daniel O'Connor, Malick Gibani, Thomas Darton, Andrew Pollard, Mark Collins, and Daniel Humphreys

Corresponding author: Daniel Humphreys (d.humphreys@sheffield.ac.uk)

Review Timeline:

Submission Date:	21st Jan 25
Editorial Decision:	17th Mar 25
Revision Received:	30th Sep 25
Editorial Decision:	27th Oct 25
Revision Received:	6th Nov 25
Accepted:	13th Nov 25

Editor: Zeljko Durdevic

Transaction Report:

17th Mar 2025

Dear Dr. Humphreys,

Thank you for the submission of your manuscript to EMBO Molecular Medicine, and please accept my apologies for the unusual delay in getting back to you. We have now received feedback from two of the three reviewers who agreed to evaluate your manuscript. As the referee #3 despite several promises will unfortunately not be able to return his/her report in a timely manner, and given that both reviewers provide very similar recommendations, we prefer to make a decision now in order to avoid further delay in the process.

As you will see from their reports pasted below, both referees recognize potential interest of the manuscript but also raise serious and partially overlapping concerns that should be addressed in a major revision. Should referee #3 provide a report, we will send it to you, with the understanding that we will not ask for an additional revision. If you would like to discuss further the points raised by the referees, I am available to do so via email or video. Let me know if you are interested in this option.

We would welcome the submission of a revised version within three months for further consideration. Please let us know if you require longer to complete the revision.

I look forward to receiving your revised manuscript.

Yours sincerely,

Zeljko Durdevic

We require:

- 1) A .docx formatted version of the manuscript text (including legends for main figures, EV figures and tables). Please make sure that the changes are highlighted to be clearly visible.
- 2) Individual production quality figure files as .eps, .tif, .jpg (one file per figure). For guidance, download the 'Figure Guide PDF': (<https://www.embopress.org/page/journal/17574684/authorguide#figureformat>).
- 3) A .docx formatted letter INCLUDING the reviewers' reports and your detailed point-by-point responses to their comments. As part of the EMBO Press transparent editorial process, the point-by-point response is part of the Review Process File (RPF), which will be published alongside your paper.
- 4) A complete author checklist, which you can download from our author guidelines (<https://www.embopress.org/page/journal/17574684/authorguide#submissionofrevisions>). Please insert information in the checklist that is also reflected in the manuscript. The completed author checklist will also be part of the RPF.

6) It is mandatory to include a 'Data Availability' section after the Materials and Methods. Before submitting your revision, primary datasets produced in this study need to be deposited in an appropriate public database, and the accession numbers and database listed under 'Data Availability'. Please remember to provide a reviewer password if the datasets are not yet public (see <https://www.embopress.org/page/journal/17574684/authorguide#dataavailability>).

12) Author contributions: You will be asked to provide CRediT (Contributor Role Taxonomy) terms in the submission system. These replace a narrative author contribution section in the manuscript.

13) A Conflict of Interest statement should be provided in the main text.

14) Every published paper now includes a 'Synopsis' to further enhance discoverability. Synopses are displayed on the journal webpage and are freely accessible to all readers. They include a short stand first (maximum of 300 characters, including space) as well as 2-5 one-sentences bullet points that summarizes the paper. Please write the bullet points to summarize the key NEW findings. They should be designed to be complementary to the abstract - i.e. not repeat the same text. We encourage inclusion of key acronyms and quantitative information (maximum of 30 words / bullet point). Please use the passive voice. Please attach these in a separate file or send them by email, we will incorporate them accordingly.

15) Include a Reagents and Tools Table as part of the Methods section, which can be downloaded from our author guidelines (<https://www.embopress.org/page/journal/17574684/authorguide#structuredmethods>)

***** Reviewer's comments *****

Referee #1 (Remarks for Author):

In this study, Scruir et al. conducted a quantitative proteomic analysis of plasma from bacteraemic individuals infected with *Salmonella Typhi* and found that toxigenic *Salmonella* infection triggers the secretion of LYZ and APOC3. Exposure to purified recombinant typhoid toxin similarly induced LYZ and APOC3 secretion in cultured intestinal cells in an ATM/ATR-mediated DNA damage response-dependent manner. Further investigation revealed that LYZ alone suppressed the secretion of key *Salmonella* SPI-1 virulence effectors, while also disrupting *Salmonella* morphology, leading to spheroplast formation. These findings suggest that toxin-induced DDR pathways activate antimicrobial responses that help control *Salmonella* bacteraemia during typhoid fever. The data generally align with the proposed claims; however, further evidence is required to fully substantiate specific mechanistic details:

1. All experiments shown in the main figures were conducted using the Caco-2 cell line, and no significant increase in LYZ expression was observed in THP-1 macrophage-like cells. Did the authors test other cell types or alternative intestinal epithelial cell lines? Additionally, the lack of a significant difference in LYZ levels between TxWT and TxHQ in THP-1 macrophages is briefly mentioned. Could this be due to a distinct DDR mechanism operating in macrophages compared to epithelial cells? Further discussion on this point would help clarify potential cell type-specific responses.
2. The authors demonstrated that treatment with the ATM/ATR inhibitor (caffeine) reduced LYZ and APOC3 expression, indicating that the expression of these genes is dependent on DDR pathways. While Figure 6 shows that adding these enzymes inhibits the secretion of T3SS effectors, it would be valuable to test whether direct inhibition of DDR pathways similarly interferes with *Salmonella* virulence factor secretion via the LYZ activity. A possible approach could involve incubating *Salmonella* with culture media from either *Salmonella*-infected cells or typhoid toxin-treated cells, followed by detection of T3SS effector protein levels. This experiment could help clarify whether DDR pathway activation induced by typhoid toxin directly modulates *Salmonella* virulence.
3. How does the typhoid toxin-induced DDR pathway trigger LYZ expression in host cells? Is this regulation mediated through interferon-stimulated genes or other inflammatory cytokines? Could it involve p53 activation, oxidative stress responses in the nucleus, or mitochondrial injury? Clarifying the signaling crosstalk between DDR and other cellular inflammatory pathways would provide deeper mechanistic insights into how typhoid toxin modulates host antimicrobial defenses. Further exploration of these interactions could help elucidate whether DDR directly regulates LYZ expression or acts through secondary immune signaling pathways.
4. Several supplementary results are based on a single experiment. To ensure reliability and statistical robustness, these experiments should be repeated at least three times independently, with appropriate statistical analyses to support the conclusions.
5. Based on the study by Chen et al. (PMID: 38555361), typhoid toxin appears to induce cellular senescence and the SASP through distinct mechanisms-nuclear DNA damage response (DDR) and mitochondrial damage, respectively. To improve clarity, it would be better to explicitly describe these mechanisms in the introduction.

Referee #2 (Comments on Novelty/Model System for Author):

In the manuscript titled "Typhoid toxin of *Salmonella Typhi* elicits host antimicrobial response during acute typhoid fever" Srour

and colleagues have used quantitative proteomic to explore the observation that human volunteers challenged with Typhoid toxin-negative strains of *S. Typhi* demonstrated significantly prolonged bacteremia via unknown mechanisms. The authors showed that human volunteers that were infected with *S. Typhi* expressing the typhoid toxin exhibited higher levels of lysozyme and apolipoprotein C3 in their blood serum than volunteers that were infected with a toxin-negative strain of *S. Typhi*. Cultured cells that were infected with recombinant typhoid toxin or infected with *S. Javiana*, which also expresses the typhoid toxin, have led to lysozyme and apolipoprotein C3 secretion and cellular DNA damage responses. Moreover, the authors showed that addition of recombinant lysozyme inhibited secretion of the T3SS effector proteins SipB and SopE. Taken together, the authors suggest that typhoid toxin-induced DNA damage responses elicit, in its turn, antimicrobial responses, which suppress *Salmonella* bacteremia during typhoid fever.

Referee #2 (Remarks for Author):

The presented results are interesting and the possible link between the activity of the typhoid toxin and antimicrobial response of the host, which is mediated by lysozyme secretion is intriguing. Nonetheless, the impact of this report can be increased by the addition of several experiments and controls.

1. To show that the impaired SipB and SopE secretion is due to the activity of lysozyme, can the authors show that they can reverse this phenotype by the addition of anti-lysozyme antibody or by addition of denatured lysozyme?
2. Is the effect of lysozyme on T3SS effectors specific or the result of a general membrane damage? Are other membrane proteins such as adhesins like SiiE being affected as well?
3. In Fig. 3 can the authors add a complementation control for the Δ cdtB *S. Javiana* strain?
4. In Fig. 5, the cell medium contain fetal calf serum. Does FCS contain lysozyme? If yes, how it may affect the results?
5. What is the regulatory cascade leading to elevated lysozyme secretion via DNA damage responses? Please discuss this point in more details.
6. Minor point: please introduce a space between numbers and units (e.g. 50 μ l or 5mM should read 50 μ l or 5 mM) as well as for *Salmonella* serovars (e.g. *S. Typhimurium* and *S. Javiana* should read *S. Typhimurium* and *S. Javiana*).

Response to Reviewers

The reviewers' comments are highlighted in italics below and have been addressed point-by-point with each point indicated by **green parentheses** in the response and, where appropriate, in the revised manuscript text. **Changes shadowed green.**

Reviewer 1

[R1_1] *'Did the authors test other cell types or alternative intestinal epithelial cell lines? Additionally, the lack of a significant difference in LYZ levels between TxWT and TxHQ in THP-1 macrophages is briefly mentioned. Could this be due to a distinct DDR mechanism operating in macrophages compared to epithelial cells? Further discussion on this point would help clarify potential cell type-specific responses.'*

We have addressed the points:

- (i) Yes, we now show that toxin-induced DDRs increase LYZ in CACO2, HCT116 and RKO human intestinal cells (**new Figure 5E**), HepG2 liver epithelial cells (**new Fig EV4B**) and THP1 monocytes (**new Fig EV4C**), which are cited on **page 10, 11**, and discussed on **page 14**.
- (ii) We clarify the effect of typhoid toxin in macrophages with new experiments on **page 11**. They address the point by showing that the toxin induces DDRs/LYZ in replicating epithelial cells and THP1 monocytes but not differentiated non-replicating THP1 macrophages (**new Fig EV4C**). This agrees with our previous study demonstrating toxin-induced replication stress that used THP1 cells as a model (Ibler et al 2019). The new data therefore make clear that when DDRs are observed so is increased LYZ, a point which is highlighted on **page 14** of the discussion.
- (iii) We replaced the immunofluorescence of THP1 macrophages (**Fig EV3**) with the immunoblot (**new Fig EV4C**), which made it easier to compare adherent THP1 macrophages and non-adherent THP1 monocytes that grow in suspension making imaging difficult.

[R1_2] *'....it would be valuable to test whether direct inhibition of DDR pathways similarly interferes with Salmonella virulence factor secretion via the LYZ activity. A possible approach could involve incubating Salmonella with culture media from either Salmonella-infected cells or typhoid toxin-treated cells, followed by detection of T3SS effector protein levels. This experiment could help clarify whether DDR pathway activation induced by typhoid toxin directly modulates Salmonella virulence.'*

We have addressed the point with new experiments, which demonstrate that LYZ activity from toxin-induced DDRs reduces *Salmonella* virulence. We now show that *Salmonella* infection is significantly reduced in TxWT-treated cells relative to TxHQ (**new Fig 4G, EV3E**), which indicates an antimicrobial response due to DDRs. Building on this we depleted LYZ using siRNA and found that LYZ inhibits intracellular infection in toxin-treated cells (**new Fig 4H, EV3F**). To support this observation, we show that LYZ can be endocytosed and localise to intracellular *Salmonella* (**new Fig 4I**), which agrees with LYZ phagocytosis into lysosomes where it counteracted *Staphylococcus aureus* (Shimada et al 2010). These results are cited on **page 8, 9**, discussed on **page 14, 15**, and included in the abstract on **page 1**.

[R1_3] *'How does the typhoid toxin-induced DDR pathway trigger LYZ expression in host cells? Is this regulation mediated through interferon-stimulated genes or other inflammatory cytokines? Could it involve p53 activation, oxidative stress responses in the nucleus, or mitochondrial injury? Clarifying the signaling crosstalk between DDR and other cellular inflammatory pathways would provide deeper mechanistic insights into how typhoid toxin modulates host antimicrobial defenses. Further exploration of these interactions could help elucidate whether DDR directly regulates LYZ expression or acts through secondary immune signaling pathways.'*

The reviewer asks how the toxin triggers LYZ expression in host cells. We have addressed this point with further experiments:

- (i) We found that increased LYZ expression coincided with p53 expression in HCT116, RKO and HepG2 cells (**new Figure 5E, EV4B**), which was supported by experiments in TP53 -/- HCT116 cells (**new Figure 6A-D**). We also observed toxin-induced LYZ expression in p53-deficient CACO2 and THP1 cells (**new Fig 5E, EV4C**) suggesting that p53-dependent LYZ expression is cell line specific and that LYZ is not a p53 effector. This is further substantiated by the loss of p53 and its effectors p21, but not LYZ, in CACO2 and THP1 treated with toxin (**new Figure 5E, EV4C**). These results are cited on **page 10, 11**, discussed on **page 14**, and included in the abstract on **page 1**.
- (ii) We found no toxin-induced interferon (IFN) response in HCT116 cells (**new Fig EV4D**). This showed that 'LYZ and p53 expression were IFN-independent', which is stated on **page 12** alongside the results.
- (iii) However, we did observe a role for oxidative stress (**new Fig 6E-6I**), particularly originating from damaged mitochondria, which agrees with recent findings demonstrating mitochondrial DNA damage by the toxin (Chen et al 2024). Inhibiting toxin-induced oxidative stress in mitochondria (e.g. using inhibitors CCCP, MitoQ) impaired expression of LYZ and p53, and its effectors p21 and GDF15 (**new Fig 6E-6G**). These results are cited on **page 12**, discussed on **page 14**, and included in the abstract on **page 1**.
- (iv) We sought to understand why a tumour suppressor like p53 might influence expression of the antimicrobial LYZ. p53 regulates gene networks that counteract oxidative stress (Sablina et al 2005, Liu & Gu, 2022). Interestingly, LYZ can reduce oxidative stress via free radical scavenging activity (Chen et al 2025, Zheng et al 2006). We found that supplementing toxin-treated cells with exogenous LYZ reduced toxin-induced DDRs (e.g. γ H2AX), as well as the p53-effector p21 (**new Fig 6E-6G**). Parallel immunofluorescence in toxin-treated cells also showed that exogenous LYZ reduced 8-oxoguanine (8-oxoG), a DNA lesion arising from oxidative stress (**new Fig 6H, 6I**). The results are cited on **page 12, 13**, discussed on **page 14**, and included in the abstract on **page 1**.

Taken together, the data suggests two principal observations: (i) that p53 regulates LYZ expression to dampen oxidative stress, and that (ii) LYZ suppresses ROS-mediated damage caused by typhoid toxin. The findings help elucidate the regulation of LYZ by DDRs addressing R1's points, which are presented in our **revised model in Fig 7** (cited **page 13**) and discussed on **page 14**.

[R1_4] *'Several supplementary results are based on a single experiment. To ensure reliability and statistical robustness, these experiments should be repeated at least three times independently, with appropriate statistical analyses to support the conclusions.'*

The highlighted experiments in the supplementary results have been repeated and are supported by appropriate statistical analyses, which are presented in **revised Fig EV1B-D** (cited **page 6, 7**), **EV2A, 2B, 2D, 2E** (cited **page 7, 8**), and **EV3E** (cited **page 10**).

[R1_5] *'Based on the study by Chen et al. (PMID: 38555361), typhoid toxin appears to induce cellular senescence and the SASP through distinct mechanisms-nuclear DNA damage response (DDR) and mitochondrial damage, respectively. To improve clarity, it would be better to explicitly describe these mechanisms in the introduction.'*

We had cited Chen et al 2024 in the introduction but now strengthen the text on **page 2** by stating that toxin-induced damage to both nuclear and mitochondrial DNA leads to senescence. In addition, our new data shows that senescence responses are suppressed by inhibitors of mitochondrial oxidative stress due to typhoid toxin, which agrees with Chen et al and has led to further citations of their work in the results on **page 12** and discussion on **page 14**.

Reviewer 2

[R2_1] 'To show that the impaired SipB and SopE secretion is due to the activity of lysozyme, can the authors show that they can reverse this phenotype by the addition of anti-lysozyme antibody or by addition of denatured lysozyme?'

We now show that *Salmonella* effector secretion was restored when we used denatured lysozyme (**new Fig 4E, 4F**), which is cited on **page 9**.

[R2_2] 'Is the effect of lysozyme on T3SS effectors specific or the result of a general membrane damage? Are other membrane proteins such as adhesins like SiiE being affected as well?'

We have addressed the point with new experiments:

- (i) Regarding membrane damage, mCherry was retained inside LYZ-treated *Salmonella* while mCherry was released by LYZ- and LFN-treated *Salmonella* in combination (**Figure 3G**). This suggests the effect of LYZ alone on T3SS is not due to membrane damage, which is now stated for clarity on **page 8**. As additional controls, we immunoblotted LYZ-treated *Salmonella* with antibodies to outer membrane proteins InvG and TolC (**revised Figure 4A**), which were retained in the cell lysates rather than being released into the supernatants due to membrane damage (**cited page 9**). These observations are discussed on **page 14** in the results section.
- (ii) We obtained the antibodies to the T1SS substrate SiiE from Prof. Michael Hensel in Germany but our attempts to detect SiiE were unsuccessful (stated on **page 9**). Thus, we tested flagella, a substrate of a T3SS distinct from the SPI-1 T3SS, which is exported to the cell surface. We found that export of FliC/flagellin was not affected by LYZ (**new Figure EV3D**), which is cited on **page 9**.

Taken together, the results suggest LYZ has effects that are specific to virulence effector T3SSs, which are independent of membrane damage. This conclusion is now in our discussion on **page 14**.

[R2_3] 'In Fig. 3 can the authors add a complementation control for the Δ cdtB *S. Javiana* strain?'

The control experiment has been performed and the results shown in **revised Fig 2G, 2H**, cited on **page 6**.

[R2_4] 'In Fig. 5, the cell medium contain fetal calf serum. Does FCS contain lysozyme? If yes, how it may affect the results?'

We have addressed the point with new experiments.

- (i) Yes. We found that TxWT induces secretion of ~280ng/ml LYZ from cultured CACO2 cells (**revised Figure 3C**). We also found that the supernatant of TxHQ-treated cells contained ~140ng/ml LYZ (**revised Figure 3C**), which correlates with untreated cells and FBS alone (**new Figure EV2A**). Thus, serum contains LYZ, which is now stated clearly on **page 7**. We correlate this with new experiments showing that LYZ was found in the plasma of uninfected participants, though at ~100-fold higher concentrations, e.g. ~25 μ g/ml (**new Figure EV2B**), cited on **page 7**. This aligns with our immunoblotting experiments demonstrating that 10 μ g/ml LYZ was sufficient to significantly inhibit the secretion of SipB (**revised Figure 4E, 4F**), cited on **page 9**.
- (ii) We reasoned that LYZ siRNA would reduce LYZ expression in cells but not effect LYZ already present in FBS, which would provide an internal control and enable us to directly test the significance of toxin-induced LYZ responses only. Relative to toxin-treated cells transfected with non-targeting siRNA, we found that LYZ siRNA had no effect on *Salmonella* invasion (**new Figure 4H, EV3F**), which agrees with our immunoblots showing that ~280ng/ml LYZ would not be sufficient to inhibit the SPI-1 T3SS as μ g LYZ concentrations are required (**revised Figure 4E, 4F**). However, we did find that LYZ siRNA inhibited intracellular *Salmonella* replication

(**new Figure 4H**), cited **page 10**. In support of this, we fluorescently labelled LYZ, which was endocytosed and localised with intracellular *Salmonella* (**new Figure 4I**). This agrees with LYZ inside phagosomes counteracting *Staphylococcus aureus* (Shimada et al 2010).

In summary, yes, FBS contains LYZ and the toxin-dependent increase in LYZ production inhibited intracellular *Salmonella* replication. The findings address the point and are cited on **page 7, 9 and 10**, discussed on **page 15**, and included in the abstract on **page 1**.

[R2_5] *'What is the regulatory cascade leading to elevated lysozyme secretion via DNA damage responses? Please discuss this point in more details.'*

Point overlaps with **[R1_3]**. We have addressed the point with new experiments.

- (iii) We found that toxin-mediated LYZ expression correlated with p53 activation (**new Figure 5E, EV4B**), which was confirmed using HCT116 TP53 knock out cells (**new Figure 6A-D**), cited on **page 11, 12**. We also observed that LYZ expression was mediated in p53-deficient CACO2 and THP1 cells (**new Figure 5E, EV4C**). Thus, LYZ expression is regulated by p53-dependent and -independent DDRs, which is discussed on **page 14**.
- (iv) We further test the signalling cascade leading to LYZ production. We observed that toxin-induced LYZ expression was independent of interferon responses (**new Fig EV4D**) but was triggered by mitochondrial oxidative stress (**new Fig 6E-6G**). Interestingly, LYZ has been observed to have free radical scavenging activity and can reduce oxidative stress (Chen et al 2025, Zheng et al 2006). Indeed, we found that supplementing toxin-treated cells with exogenous LYZ reduced toxin-induced oxidative stress as determined by immunoblotting (**new Fig 6E-6G**), and fluorescence microscopy (**new Fig 6H, 6I**). These results are cited on **pages 12, 13**, presented in the **revised model in Fig 7** (cited **page 13**), discussed on **page 14, 15**, and included in the abstract on **page 1**.

[R2_6] *'Minor point: please introduce a space between numbers and units (e.g. 50µl or 5mM should read 50 µl or 5 mM) as well as for Salmonella serovars (e.g. S.Typhimurium and S.Javiana should read S. Typhimurium and S. Javiana).'*

The text has been edited as suggested, i.e. S. Typhi, S. Javiana, S. Typhimurium. For example, in line 3 of the first paragraph in the introduction. Point overlaps with **[R3_4]**.

Reviewer 3

[R3_1] *'The authors demonstrate that typhoid toxin-induced DNA is a likely cause of the higher levels of LYS in TT+ve patients. However, while it is admittedly a difficult question to address, the authors show no compelling evidence that the minor increase in lysozyme serum levels (which appear to be <2-fold higher in patients infected with WT S. Typhi) has any role or impact in S. Typhi pathogenesis. The in vitro experiments they perform do not directly address this question at all. Further, the quantities of LYS they use for their in vitro studies (Fig 5) are ~100-fold higher than the serum concentrations they cite (1mg/ml - serum levels 10ug/ml). Based on their proteomics, we might expect a concentration more like 18ug/ml. Is the difference between 10 and 18 make a meaningful difference in vivo? Who knows? in vivo, at the systemic stage of infection, are S. Typhi even susceptible to lysozyme? In the serum, many (most?) S. Typhi are likely intracellular and thus not exposed to lys. Might extracellular bacteria be protected from Lys (and Lys+Lf) by the Vi antigen? Are there components of the serum (like LF) that enhance or reduce the effects of lys on Typhi? Does S. Typhi reprogram its gene expression in serum in a way that protects it from Lys? It is certainly not clear that LYS would be a dominant (or even a relevant) antimicrobial factor for S. Typhi in the serum. The experiments in Figure 5 are fine, but they do not really tell us anything that wasn't already pretty well established. All this being said, I was impressed by the authors' concessions along these lines in the discussion, ""we think it unlikely that lysozyme acts alone in counteracting Salmonella following DNA damage by the typhoid toxin"" - it was refreshing to see this objective comment that sacrificed hyping up their results in favour of a more realistic view.'*

The reviewer makes interconnected points, which are broken down and addressed below:

R3_1A: 'the authors show no compelling evidence that the minor increase in lysozyme serum levels (which appear to be <2-fold higher in patients infected with WT *S. Typhi*) has any role or impact in *S. Typhi* pathogenesis. The *in vitro* experiments they perform do not directly address this question at all'. Further, the quantities of LYZ they use for their *in vitro* studies (Fig 5) are ~100-fold higher than the serum concentrations they cite (1mg/ml - serum levels 10ug/ml). Based on their proteomics, we might expect a concentration more like 18ug/ml. Is the difference between 10 and 18 make a meaningful difference *in vivo*? Who knows? *in vivo*, at the systemic stage of infection, are *S. Typhi* even susceptible to lysozyme? In the serum, many (most?) *S. Typhi* are likely intracellular and thus not exposed to lys'.

- (i) We have determined the LYZ concentration in plasma in non-infected (25 µg/ml) and during typhoid fever (40 µg/ml) (**new Figure EV2B**), which correlates with our *in vitro* findings showing that 10 µg/ml LYZ inhibits T3SS-mediated secretion of SipB (**new Figure 4E, 4F**), cited on **page 7, 9**.
- (ii) *S. Typhi* in the blood is both extracellular (~37%) and intracellular (~63%) (Wain et al 1998), now cited in our discussion at the bottom of **page 14**.
- (iii) We now show that the toxin-dependent increase in LYZ is meaningful in LYZ siRNA-transfection experiments. The results show that LYZ significantly inhibits intracellular *Salmonella* infection of cultured cells relative to the concentration of LYZ already present in serum/FBS (**new Fig 4H**), cited on **page 10**. This suggests that LYZ is endocytosed and interacts with *Salmonella*, which we demonstrate in new fluorescence microscopy images (**new Fig 4I**), cited on **page 10**. This agrees with published findings showing that LYZ is present within phagosomes where it performs its antimicrobial activities against *Staphylococcus aureus* (Shimada et al 2010), now cited on **page 15**. The results are cited on **pages 9, 10**, discussed on **page 15**, and included in the abstract on **page 1**.

Taken together, the findings indicate that LYZ can be endocytosed and inhibit intracellular infections, which suggests that *S. Typhi* in the bloodstream are exposed to LYZ whether they are intracellular or extracellular. These points are made clear in our discussion on **page 14, 15**.

R3_1B: 'Might extracellular bacteria be protected from Lys (and Lys+Lf) by the Vi antigen?'

Regarding the Vi capsule, we tested T3SS function during LYZ treatment in Vi-positive and Vi-negative mutant *S. Typhi*, which revealed that Vi plays no protective role against LYZ-mediated inhibition of T3SS activity (**new Figure EV3B-C**), cited **page 9**.

R3_1C: 'Are there components of the serum (like LF) that enhance or reduce the effects of lys on *Typhi*'?

Yes, for example, defensins synergised with LYZ against *S. aureus* and *E. coli* (Chen et al 2005), and defensin concentrations are elevated in plasma during bacterial infections (Panyutich et al 1993). This point is made and the publications cited in the results on **page 7**.

R3_1D: 'Does *S. Typhi* reprogram its gene expression in serum in a way that protects it from Lys?'

This is very likely as a *Salmonella* inhibitor of LYZ called MliC/YdhA was up-regulated by *S. Typhi* within macrophages and its deletion inhibits *Salmonella* invasion and survival in macrophages (Ragland & Kriss 2017, Daigle et al 2008). We make this point in our discussion on **page 15** saying that *Salmonella* protects itself intracellularly from LYZ by expressing MliC.

[R3_2] 'Cell envelope damage broadly impacts many aspects of bacterial physiology. It is not surprising that the activity of a T3SS falls into this category. But this whole line of questioning (Fig 6) does not seem to be based on a logical progression of sound scientific reasoning. Out of the many ways cell wall damage might reduce *S. Typhi* fitness during systemic infections, why choose to examine SPI-1? Is the SPI-1 T3SS (active cell invasion) in the serum of patients relevant for

systemic infections? To my knowledge, this isn't clear. SPI-1 plays important roles for Typhi invading non-phagocytic cells in the intestines and possibly the gallbladder. The idea that this small increase in LYS serum levels might trigger exactly the right level of cellular damage to be specifically reducing activity of the SPI-1 T3SS (which might not be doing anything relevant here anyway)...it just isn't the most plausible impact for LYS...and that's assuming LYS is even relevant at all (see point above). Jumping from LYS to assessing SPI-1 feels completely arbitrary. If LYS is doing enough damage to the cell wall to stop a T3SS from functioning, there are presumably more direct impacts that are much more likely to be responsible for impacting bacteraemia in typhoid fever patients.

I don't believe the T3SS line of examination makes sense at face value, so I hesitate to get into the weeds here, but T3SS are programmed to secrete effectors upon contact with host cells. Under inducing conditions and when large numbers of bacteria are cultured, you can detect some unprimed secretion into the growth media - this is what they assess here. How meaningful is that? Does LYS reduce cell invasion? Is this because of T3SS inhibition, or some other envelope damage effect? There are other issues with this data as well; a big one is that there is no quantification/statistical significance here. Most of the experiments are done at 1mg/ml lysozyme (100x the stated serum levels). The lone panel that shows data for lower LYS concentration ranges, the authors used *S. Javiana* (why not Typhi???) and the data point that is most relevant - 10ug/ml - is oddly absent (data jumps from 100ug/ml to 1ug/ml). But again, I suggest removing this section altogether, so I am not recommending they complete more experiments to address these issues.'

R3 questions the suitability of investigating the SPI-1 T3SS (R3_2A), whether the effect of LYZ is due to cell wall damage (R3_2B), the lack of a 10µg/ml LYZ treatment concentration in our blots (R3_2C), and our use of *S.Javiana* rather than *S.Typhi* (R3_2D). The comments are addressed below:

R3_2A: This point overlaps with R3_1A. Yes, the SPI-1 T3SS is expressed by extracellular *Salmonella* but also has a role in intracellular infections (McGhie et al 2009), which is now made clear for the reader on paragraph 2 of the introduction on **page 3**, and the discussion on **page 15**. We further address the point by citing evidence from Wain et al 1998 highlighting that 37% of *Salmonella* are extracellular in the human bloodstream where *Salmonella* would likely express the SPI-1 T3SS (cited in the discussion on **page 14**). However, we acknowledge that 63% of *Salmonella* are intracellular and studying this phase of infection makes logical sense: we make this clear in the results on **page 10** before showing that LYZ inhibits intracellular infections and that LYZ can localise to intracellular *Salmonella* (**new Figure 4H, 4I, EV3F**). The findings correlates with toxin-induced suppression of bacteraemia in human infection challenge studies (Gibani et al 2019) and is discussed on **page 15**, and included in the abstract on **page 1**.

R3_2B: We have examined whether the LYZ effect is specific to T3SS or due to membrane damage (point addressed in R2_2). We find that LYZ and LFN cause membrane damage (**Figure 3F, 3G**), which was not observed with LYZ alone and appears to effect the T3SS specifically via an unknown mechanism (**Figure 3G, 4A, EV3D**). Results cited on **pages 8, 9**, and discussed on **page 15**.

R3_2C: We now show *Salmonella* treatment with 10 µg/ml LYZ (**new Figure 4E, 4F**), which shows that 10µg/ml LYZ significantly inhibits T3SS activity and the results are cited on **page 9**.

R3_2D: *S. Javiana* is a hazard group 2 pathogen while *S. Typhi* is hazard group 3 in Europe and also comes under schedule 5 of the anti-terrorist act 2001 in the UK. Thus, wild-type *S. Javiana* and attenuated hazard group 2 *S. Typhi* were used for biosecurity reasons, and *S.Javiana* used as a preference as it is a wild-type strain. This point is now emphasised in the results on **page 6**.

[R3_3] *'There was an odd lack of replicates/stats/quantification for some panels. Figs 4C/4D - no stats given, n=1 for one panel and n=2 for the other. Should expand to at least n=3 and get p values for this. Similarly, n=1 in Fig 5c, Fig 5B shows no significance analysis. Figure 6 (western blots for secretion assays) shows no significance, no quantification.'*

We have addressed the points by performing at least 3 independent experiments and supporting the data with statistical analysis in revised Figure 5B, 5C (**now revised Fig 3B, 3C**), and the immunoblots in Figure 6 (now **revised Fig 4**) have supporting quantification in **new Fig 4B, 4C, 4F**. The new data support our conclusions, demonstrate reproducibility, and are cited on **pages 7, 9**.

[R3_4] *'Throughout the entire manuscript S. Typhi is written without a space "S.Typhi". Not sure what's going on here...a formatting issue?'*

The text has been edited as suggested, i.e. S. Typhi, S. Javiana, S. Typhimurium. For example, in line 3 of the introduction on **page 2**. Point overlaps with **[R2_6]**.

[R3_5] *'I had to read the final paragraph of the introduction a few times to follow the logic. To the reader who is not yet familiar with where this paper is going, this paragraph is hard to digest. I feel that this paragraph could be re-written to clarify - this paragraph jumps around a bit without clear transitions/connections between ideas.'*

We have changed the text in this paragraph on **page 4** of the introduction to make it more clear for the reader. Firstly, we have added the sentence *'but how toxin-induced DDRs might counteract infection is not known'* to highlight the gap in knowledge between toxin-induced DDRs and host defence mechanisms. Secondly, we have removed text relating to NTS models in mice so the paragraph continues instead to ask broader questions relating to DDRs and host defences, which are more pertinent to the manuscript.

[R3_6] *'The start of the results section should be more explicit that the proteomics was done using the same patient samples collected and analyzed previously in the 2019 paper by Gibani et al.'*

We have modified the text by adding *'in samples harvested by Gibani and colleagues'* in the first paragraph of the results on **page 5**. This provides additional clarification as to the source of the plasma samples.

[R3_7] *'Results paragraph #1 - typo - states depletion was done to reduce the dynamic range...I imagine this is supposed to read "enhance" the dynamic range.'*

We have made the edit in the results on **page 5**.

[R3_8] *'When ApoC3 is first introduced, full name is given. When LYS is introduced (top of pg. 5), should give name (lysozyme) as well.'*

'Lysozyme (LYZ)' was already introduced at the bottom of **page 5**.

[R3_9] *'There are two versions of typhoid toxin with different B subunits. This should be mentioned somewhere and it should be clarified that this study only focused on the PltB toxin. This isn't a problem given the nature of their results/experiments, but it should be pointed out.'*

We now strengthen the introduction by stating that typhoid toxin can be assembled with PltC in place of PltB on **page 3**. We also highlight that typhoid toxin is expressed by different serovars of *Salmonella*, also on **page 3**.

[R3_10] *'It seems to me that Figs 2/3 could be merged into one figure. They are thematically the same and there looks to be space to do this'.*

We have merged Figure 2 and 3, as suggested, which generates a **revised Figure 2**, cited on **page 6**.

[R3_11] *'Borderline "Major issue" - data organization is a bit odd between Figs 4/5. Why aren't panels 5A-5C in Fig 4? Figure 4 looks at toxin/DDR-mediated increase in Lys - that's where this data seems to belong. Fig 5 is about the effects of LYS on bacterial cells...not nearly as good a fit here'.*

The figures have been reorganised to integrate the new data and to improve the logical flow of the manuscript. As suggested, the former Figure 5A–5C now appears earlier. We believe this restructuring clarifies the progression of the study, which is summarised below:

- (i) **Revised Figure 2:** Former Figures 2 and 3 have been merged to validate toxin-induced APOC3 expression.
- (ii) **New Figure 3A–3C:** The previous Figure 5A–5C have been moved earlier. These panels validate toxin-induced LYZ expression in cultured cells, paralleling the approach taken for APOC3 in Figure 2. We consider it logical to establish LYZ induction (Figure 3A–3C) before assessing its functional effects on *Salmonella* in the subsequent panels (Figure 3D–3H) and during infection (Figure 4).
- (iii) **Figures 5 and 6:** We then analyse the signalling cascades involving LYZ, DDRs, and oxidative stress. Their placement later in the manuscript reflects the interest from R1/R2 in LYZ regulation and allows us to highlight the new mechanistic insights generated in response to their comments

[R3_12] *'The statement "LFN was found in all participants by proteomics indicating that LYZ and LFN work in combination to suppress Salmonella in response to typhoid toxin." is a massive leap of faith and is not sound. This should be replaced by something like: "LFN was found in all participants by proteomics, SUGGESTING THE POSSIBILITY that LYZ and LFN COULD work in combination to suppress Salmonella in response to typhoid toxin."'*

We have made the text edit in the results section on **page 7**.

27th Oct 2025

Dear Dr. Humphreys,

Thank you for the submission of your revised manuscript to EMBO Molecular Medicine. I am pleased to inform you that we will be able to accept your manuscript pending the following final amendments:

- 1) Authors: Please define the corresponding author on the title page and add the email address.
 - 2) Figures: We note that some images are reused. Figure 3G (untreated and LYZ) are reused in Figure EV2C. Please cite in the respective figure legend every reused image/panel.
 - 3) Please address all comments suggested by our data editors listed below:
 - o Data availability statement:
 1. Please note that the specific URL for PXD058381 dataset is not provided in the data availability statement.
 - o Figure legends:
 1. Please note that the exact p values are not provided in the legends of figures 2B, C, D, E, G, H; 3B, C, E, H; 4B, C, F, G, H; 5B, C; 6B, C, F, G, I; EV1 B, C; EV2 B, E
 2. Please note that information related to n is missing in the legend of figure 1D.
 - Rename Summary to Abstract.
 - Add up to 5 keywords.
 - In Methods, provide the statement that informed consent was obtained from all human subjects and confirm that the experiments conformed to the principles set out in the WMA Declaration of Helsinki and the Department of Health and Human Services Belmont Report.
 - Please remove "Figures" and "Availability of new reagents".
 - Indicate in legends exact n and exact p values, not a range, along with the statistical test used. To keep the figures "clear" some authors found providing an Appendix table Sx with all exact p-values preferable. You are welcome to do this if you want to.
 - Please remove Reagents and Tools Table and uploaded it as a separate file. Structured Methods section includes Reagents and Tools Table followed by a Methods and Protocols section. More information on how to adhere to this format as well as downloadable templates (.docx) for the Reagents and Tools Table can be found in our author guidelines: <https://www.embopress.org/page/journal/17574684/authorguide#structuredmethods>
- An example of a paper with Structured Methods can be found here: <https://www.embopress.org/doi/full/10.1038/s44320-024-00037-6#sec-4>
- Rename "Conflict of Interest statement" to "Disclosure Statement & Competing Interests" and place it after the "Acknowledgements". We updated our journal's competing interests policy in January 2022 and request authors to consider both actual and perceived competing interests. Please review the policy <https://www.embopress.org/competing-interests> and update your competing interests if necessary.
- Author contributions: Please remove it from the manuscript and specify author contributions in our submission system. CRediT has replaced the traditional author contributions section because it offers a systematic machine-readable author contributions format that allows for more effective research assessment. You are encouraged to use the free text boxes beneath each contributing author's name to add specific details on the author's contribution. More information is available in our guide to authors: <https://www.embopress.org/page/journal/17574684/authorguide#authorshipguidelines>
- In data availability statement remove the reference to BioStudies as you have already submitted source data. Please use the following format to report the accession number of your data:

[data type]: [full name of the resource] [accession number/identifier] ([doi or URL or identifiers.org/DATABASE:ACCESSION])

Please check "Author Guidelines" for more information.

<https://www.embopress.org/page/journal/17574684/authorguide#availabilityofpublishedmaterial>

4) Funding: Please make sure that information about all sources of funding are complete in both our submission system and in the manuscript. Currently, MR/R005982/1 is missing in the manuscript text and MR/N013840/1 is missing in our submission system. Please correct.

5) Datasets: Please correct file names to Dataset EV1 and EV2.

6) Synopsis:

- Synopsis image: Please provide a visual abstract as a high-resolution jpeg file 550 px-wide x 300-600 pixels high to illustrate your article.

7) As part of the EMBO Publications transparent editorial process initiative (see our Editorial at

<http://embomolmed.embopress.org/content/2/9/329>), EMBO Molecular Medicine will publish online a Review Process File (RPF) to accompany accepted manuscripts. This file will be published in conjunction with your paper and will include the anonymous referee reports, your point-by-point response and all pertinent correspondence relating to the manuscript. Let us know whether

you agree with the publication of the RPF and as here, if you want to remove or not any figures from it prior to publication. Please note that the Authors checklist will be published at the end of the RPF.

8) Please provide a point-by-point letter INCLUDING my comments as well as the reviewer's reports and your detailed responses (as Word file).

I look forward to reading a new revised version of your manuscript as soon as possible.

Yours sincerely,

Zeljko Durdevic

Zeljko Durdevic
Senior Editor
EMBO Molecular Medicine

*** Instructions to submit your revised manuscript ***

- 1) a .docx formatted version of the manuscript text (including Figure legends and tables)
- 2) Separate figure files*
- 3) supplemental information as Expanded View and/or Appendix. Please carefully check the authors guidelines for formatting Expanded view and Appendix figures and tables at <https://www.embopress.org/page/journal/17574684/authorguide#expandedview>
- 4) a letter INCLUDING the reviewer's reports and your detailed responses to their comments (as Word file).
- 5) The paper explained: EMBO Molecular Medicine articles are accompanied by a summary of the articles to emphasize the major findings in the paper and their medical implications for the non-specialist reader. Please provide a draft summary of your article highlighting
 - the medical issue you are addressing,
 - the results obtained and
 - their clinical impact.This may be edited to ensure that readers understand the significance and context of the research. Please refer to any of our published articles for an example.
- 6) Author contributions: the contribution of every author must be detailed in a separate section.
- 7) EMBO Molecular Medicine now requires a complete author checklist (<https://www.embopress.org/page/journal/17574684/authorguide>) to be submitted with all revised manuscripts. Please use the checklist as guideline for the sort of information we need WITHIN the manuscript. The checklist should only be filled with page numbers where the information can be found. This is particularly important for animal reporting, antibody dilutions (missing) and exact values and n that should be indicated instead of a range.

8) Every published paper now includes a 'Synopsis' to further enhance discoverability. Synopses are displayed on the journal webpage and are freely accessible to all readers. They include a short stand first (maximum of 300 characters, including space) as well as 2-5 one sentence bullet points that summarise the paper. Please write the bullet points to summarise the key NEW findings. They should be designed to be complementary to the abstract - i.e. not repeat the same text. We encourage inclusion of key acronyms and quantitative information (maximum of 30 words / bullet point). Please use the passive voice. Please attach these in a separate file or send them by email, we will incorporate them accordingly.

You are also welcome to suggest a striking image or visual abstract to illustrate your article. If you do please provide a jpeg file 550 px-wide x 300-600px high.

9) A Conflict of Interest statement should be provided in the main text

10) Please note that we now mandate that all corresponding authors list an ORCID digital identifier. This takes <90 seconds to complete. We encourage all authors to supply an ORCID identifier, which will be linked to their name for unambiguous name identification.

Currently, our records indicate that the ORCID for your account is 0000-0002-1038-2538.

Link Not Available

11) Include a Reagents and Tools Table as part of the Methods section, which can be downloaded from our author guidelines (<https://www.embopress.org/page/journal/17574684/authorguide#structuredmethods>)

Photos 400-800 DPI

*Additional important information regarding figures and illustrations can be found at <https://bit.ly/EMBOPressFigurePreparationGuideline>. See also figure legend preparation guidelines: <https://www.embopress.org/page/journal/17574684/authorguide#figureformat>

***** Reviewer's comments *****

Referee #1 (Remarks for Author):

The revised manuscript has successfully addressed my previous concerns. The authors have provided sufficient additional data and clarification, which substantially strengthen the study.

The authors addressed the remaining editorial issues.

13th Nov 2025

Dear Dr. Humphreys,

We are pleased to inform you that your manuscript is accepted for publication and is now being sent to our publisher to be included in the next available issue of EMBO Molecular Medicine.

Zeljko Durdevic
Senior Editor
EMBO Molecular Medicine
